# Antarctic surface climate and surface mass balance in the Community Earth System Model version 2 during the satellite era and into the future (1979-2100)

Devon Dunmire[1], Jan T. M. Lenaerts[1], Rajashree Tri Datta[1], and Tessa Gorte[1]

[1]Department of Atmospheric and Oceanic Sciences, University of Colorado Boulder, USA

**Correspondence:** Devon Dunmire (devon.dunmire@colorado.edu)

**Abstract.** Earth System Models (ESMs) allow us to explore minimally-observed components of the Antarctic Ice Sheet (AIS) climate system, both historically and under future climate change scenarios. Here, we present and analyze surface climate output from the most recent version of the National Center for Atmospheric Research's ESM: the Community Earth System Model version 2 (CESM2). We compare AIS surface climate and surface mass balance (SMB) trends as simulated by CESM2 with reanalysis and regional climate models and observations. We find that CESM2 substantially better represents the mean state AIS near-surface temperature, wind speed, and surface melt compared with its predecessor, CESM1. This improvement likely results from the inclusion of new cloud microphysical parameterizations and changes made to the snow model component. However, we also find that grounded CESM2 SMB ($2269 \pm 100$ Gt yr$^{-1}$) is significantly higher than all other products used in this study and that both temperature and precipitation are increasing across the AIS during the historical period, a trend that cannot be reconciled with observations. This study provides a comprehensive analysis of the strengths and weaknesses of the representation of AIS surface climate in CESM2, work that will be especially useful in preparation for CESM3 which plans to incorporate a coupled ice sheet model that interacts with the ocean and atmosphere.

## 1 Introduction

The Antarctic Ice Sheet (AIS) is the largest freshwater body on Earth, storing enough ice to raise the global mean sea level by 58.3 m if melted entirely (Church et al., 2013). The mass balance of the AIS is equivalent to the difference between surface mass balance (SMB), which is *precipitation - evaporation/sublimation - runoff*, and ice discharge, or the ice flux across the grounding line. Observations indicate that the AIS has been losing mass since the late 1970s, implying that ice discharge has exceeded mass gain due to SMB. AIS mass loss has increased from $40 \pm 9$ Gt yr$^{-1}$ in 1979–1990 to $252 \pm 26$ Gt yr$^{-1}$ in 2009–2017 (Rignot et al., 2019). This mass loss is focused in the Amundsen Sea sector and the Antarctic Peninsula, combined accounting for 81% of the total AIS mass loss between 2003 and 2013 (Velicogna et al., 2014). Ice shelves in the Amundsen and Bellingshausen sea regions are thinning in large part due to increased basal melting (Pritchard and others, 2012), a process that reduces the buttressing effect of ice shelves and leads to increased ice discharge (Rignot et al., 2019; Milillo et al., 2022).

SMB is important for AIS mass balance because, when increasing, it can counteract increased discharge and mitigate the ice sheet's contribution to sea level rise. Precipitation dominates the AIS SMB signal and is variable from year to year, impacted by

modes of variability (Hansen et al., 2021; Marshall et al., 2017), stratospheric ozone depletion (Lenaerts et al., 2018; Chemke et al., 2020; Schneider et al., 2020), and increasing greenhouse gas emissions (Palerme et al., 2017). Historical increases in AIS SMB indicate that some of this mass loss mitigation may already be happening (Medley and Thomas, 2019); however, uncertainty remains as to what extent this will continue in the future (Lenaerts et al., 2016; Gorte et al., 2020).

While increasing snowfall is important for mitigating AIS mass loss due to increased discharge, other surface processes,
such as surface melt and rain, will also play a growing role in the future of the AIS. Surface meltwater impacts ice shelves, which surround 75% of the Antarctic coastline, and provide a buffer from the inland flow of ice to the ocean (Fürst et al., 2016). Surface meltwater ponding can lead to hydrofracture (Banwell et al., 2019; Dunmire et al., 2020), or the rapid vertical drainage of meltwater, a process which may drive ice-shelf instability and break-up (Gilbert and Kittel, 2021; Robel and Banwell, 2019; Banwell et al., 2013; Scambos et al., 2009).

Because of Antarctica's remoteness, in-situ observations are spatially and temporally sparse, limiting our understanding of how the surface climate and SMB are changing. Accordingly, we use additional products to assess the AIS surface climate, each with its own set of advantages and disadvantages. Satellite remote sensing products provide observations across the ice sheet but are not continuous, only exist for a short period of time, and cannot directly measure SMB (and indirect remote-sensing measurements of SMB come with large uncertainties). Reanalysis models, such as ERA5 and MERRA-2, and SMB
reconstructions, such as that from Medley & Thomas (2019), approximate observations as best as possible, but only exist for the historical period. Regional climate models (RCMs) can be useful tools for analysing AIS surface climate and surface mass balance (Mottram et al., 2021) but are expensive to run and require lateral boundary forcing from other global products. These limitations highlight the important gap that Earth System Models (ESMs) fill. ESMs represent many components of the climate system, allowing for the analysis of climate interactions, feedbacks, and internal variability. Further, ESMs are integrated in the
most recent Coupled Model Intercomparison Project (CMIP6, Eyring et al., 2016), which provides future climate projections under a combination of different radiative forcing (RCP) and socioeconomic pathways (SSPs), and are used as forcing for ice sheet models (e.g. Seroussi et al., 2020).

The spread of how well various ESMs within CMIP6 capture AIS SMB is very large. CMIP6 modeled annual SMB values between 1950 and 2000 range between 1525 and 3378 $\mathrm{Gt\ yr^{-1}}$, with a mean of 2127 $\mathrm{Gt\ yr^{-1}}$ (Gorte et al., 2020). To better
understand this spread in CMIP6 models and help inform future decisions regarding ice sheet model forcing, ESM evaluation exercises are important. Here, we present and investigate output from the most recent version of National Center for Atmospheric Research's ESM: the Community Earth System Model version 2 (CESM2, Danabasoglu et al., 2020). We compare this model with its predecessor (CESM1) to highlight model improvements. We also compare CESM2 surface climate output with observations from autonomous weather stations (AWSs) across the AIS, satellite observations, and output from reanaly-
sis models and an SMB reconstruction to emphasize potential areas of improvement for the next model version. Finally, we explore historical and future trends in the model, relating to surface mass balance.

## 2 Methods and Data

### 2.1 Community Earth System Model

#### 2.1.1 CESM2

Here, we analyzed output from the Community Earth system Model Version 2 (CESM2), the National Center for Atmospheric Research's Earth System Model. CESM2 is an open-source community model consisting of fully coupled ocean, atmosphere, land, sea-ice, land-ice, river, and wave models at $\sim1°$ horizontal resolution. In this study, we analyzed model output from the CMIP6 archive, which includes 11 ensemble members covering the historical period (1850-2015), as well as 3 ensemble members covering the remainder of the 21st century (2015-2100) following three different future socioeconomic pathways (SSPs), SSP1-2.6, SSP3-7.0, and SSP5-8.5. CESM2 has multiple elevation classes active over Antarctica. Because the downscaling does not change the grid cell integrated mass or energy fluxes, CESM2 is not coupled to an ice sheet model over the AIS, and most atmospheric variables are not downscaled, we present our results on the native CESM2 grid.

We used near-surface air temperature, near-surface wind speed, incoming longwave radiation, incoming shortwave radiation, latent heat flux, sensible heat flux, sea level pressure, and geopotential height output from the atmosphere model, the Community Atmosphere Model Version 6 (CAM6). Runoff, solid and liquid precipitation, evaporation/sublimation, and melt output were obtained from the land model, the Community Land Model Version 5 (CLM5, Lawrence et al., 2019). For comparing the CESM2 mean and uncertainty of these output variables to other products we calculated the 11-member ensemble average mean and standard deviation.

We also compared CESM2 Antarctic SMB output (as part of CMIP6) with the 100-member CESM2 Large Ensemble Project (CESM2-LENS, Rodgers et al., 2021). However, we used the 11-member CESM2 output for the majority of the analysis in this work because it contains output from 3 different future scenarios where-as CESM2-LENS only contains output from SSP3-7.0.

#### 2.1.2 Model differences from CESM1

We evaluated the impact of three major changes that were made to CESM2's predecessor, the CESM1 Large Ensemble (CESM1-LENS, CESM1 hereafter, Kay et al. (2015)). First, the inclusion of new cloud microphysical parameterizations such as ice nucleation and prognostic precipitation allow for a better representation of clouds in polar regions and therefore led to improved modeled air temperatures, incoming longwave and shortwave radiation, and surface melting (Lenaerts et al., 2020). Secondly, changes made to the snow model over land, such as implementing new parameterizations for fresh snow density, destructive metamorphism, and compaction by overburden pressure and wind redistribution and allowing for a deeper firn layer have improved the representation of perennial snow in polar regions and have implications for simulated surface meltwater production, refreezing, and runoff (van Kampenhout et al., 2017). Thirdly, CESM2 includes a new parameterization for boundary layer form drag (Beljaars et al., 2004), which has been shown to improve the representation of orographic precipitation, near-surface wind, and turbulent heat and moisture fluxes over Greenland (van Kampenhout et al., 2020).

## 2.2 Other modeling and observational Products

To evaluate CESM2, we compared model output to in-situ observations, remote sensing products, atmospheric reanalysis models, RCMs, and an SMB reconstruction product, described below.

### 2.2.1 In-situ observations

We used near-surface temperature and wind speed observations from a collection of 133 automatic weather stations (AWSs) across the Antarctic Ice Sheet (Gossart et al., 2019). This collection was downsized to only include stations that contained 10 or more full years of temperature or wind speed data. Ultimately, we used near-surface temperature observations from 116 different AWSs and near-surface wind speed observations from 96 different AWSs.

### 2.2.2 Remote sensing products

We used melt observations which were empirically derived from radar backscatter from the QuikSCAT (QSCAT) satellite (Trusel et al., 2013). QSCAT observations are available at a horizontal scale of 27.2 $km^2$ and were upscaled to the same grid as CESM2 using bilinear regridding.

### 2.2.3 Atmospheric reanalysis, RCM, and SMB reconstruction products

We compared CESM2 AIS SMB to a collection of other atmospheric reanalysis, RCM, and SMB reconstruction products. In all modeling products, SMB is approximated by *precipitation - evaporation/sublimation - runoff*. We used atmospheric reanalysis product ERA5 (Hersbach et al., 2020) which is produced by the European Center for Medium-range Weather Forecasts (ECMWF) and assimilates observations at a horizontal resolution of $\sim$30 $km^2$. For RCMS, we used output from the latest versions of RACMO2.3, which is forced with ERA-Interim (van Wessem et al., 2017), and MAR (version 3.11), which is forced with ERA5 (Kittel et al., 2021). The SMB reconstruction is a product generated by Medley and Thomas (2019) which provides AIS SMB from 1801-2000 by synthesizing ice core records with reanalysis products. In this study we used the MERRA-2 based SMB reconstruction as it most closely resembles observations (Medley and Thomas, 2019). We will refer to this product as "the MT2019 reconstruction" and we used the SMB error provided by Medley and Thomas (2019) as the variability for this dataset.

We also compared the CESM2 trend in near-surface temperature and precipitation from 1979-2015 with that from ERA5. We used ERA5 for this comparison because (a) it is the latest reanalysis product, with updated model physics and the highest horizontal resolution, and (b) has similar near-surface temperature and precipitation trends to the RCMS used in this study (Fig. A1, A2). The ERA5 near-surface temperature trend is also consistent with observations (Zhu et al., 2021).

## 2.3 Model AIS masks

For area-integrated quantities we used the Zwally et al. (2012) AIS mask which has been re-gridded for all of the modeling products used in this study. The resulting grounded AIS areas from these models are as follows: 12,043,565 $km^2$ for CESM1

and CESM2, 12,059,084 km$^2$ for ERA5, 12,063,497 km$^2$ for RACMO2.3, 12,154,338 km$^2$ for MAR, and 12,028,208 km$^2$ for the MT2019 reconstruction. The resulting ice shelf areas from these models are: 1,738,581 km$^2$ for CESM1 and CESM2, 1,755,916 km$^2$ for ERA5, 1,734,991 km$^2$ for RACMO2.3, and 1,749,205 km$^2$ for MAR. Ice shelves were not included in the MT2019 reconstruction.

## 2.4 SMB component comparison

To compare the relative importance of each SMB component during different time periods and from different model output we divided each component by the sum of the magnitude of all components, which we call the "SMB signal" throughout Section 3. For example, the contribution of runoff to the SMB signal was determined by:

$runoff_{contribution} = \frac{|runoff|}{|precipitation|+|evaporation/sublimation|+|runoff|}$, where $precipitation$ is the sum of both solid (snowfall) and liquid (rainfall) precipitation. This creates a standardized method to compare the relative importance of each SMB component among different models and scenarios.

## 3 Results

### 3.1 Near-surface temperature

Modeled annual AIS near-surface (2 m) air temperature in CESM2 between 1979 and 2015 ranges from -52 °C in the high-elevation interior to -7 °C along the coast (Fig.1a). Average annual near-surface air temperature in CESM2 is 2.86 ± 0.66 °C warmer than in CESM1 (Fig. 1b), with the largest temperature increase between model versions during the austral winter season (Fig. 1d). However, modeled near-surface air temperature in CESM2 is still generally underestimated relative to observations across the AIS (Fig. 1c). The average annual temperature bias between CESM2 and observations at 116 different AWSs is -2.98 °C, an improvement from -5.18 °C in CESM1. Similar to CESM1, near-surface air temperature in CESM2 is positively biased in the high elevation interior and negatively biased along the coast (Lenaerts et al., 2016). The bias between CESM2 and AWS observations at sites with an elevation > 2000 m is +0.82 °C, which is significantly different (p < 0.05) from the -3.59 °C average bias at sites with an elevation < 2000 m. There are relatively more AWSs at low elevation sites, which leads to the overall average negative bias between CESM models and AWS observations.

Both models show similar seasonality in their bias with respect to AWS observations, with better agreement during the austral summer and the highest bias during the austral winter (Fig. 1d), which is likely due to an underestimation of inversion strength, a common issue for climate models (Vignon et al., 2018).

A likely reason for the improvement in modeled near-surface air temperature in CESM2 compared to CESM1 is the enhanced cloud liquid water over high latitudes (Lenaerts et al., 2020). Liquid-containing clouds enhance shortwave radiation blocking, but are efficient absorbers of longwave radiation, leading to a decrease in incoming shortwave radiation (Fig. 2b) and an increase in incoming longwave radiation (Fig. 2e) across the entire AIS in CESM2, compared with CESM1. In polar regions, typically the longwave effect of clouds dominates because (1) incoming shortwave radiation only plays a role during the

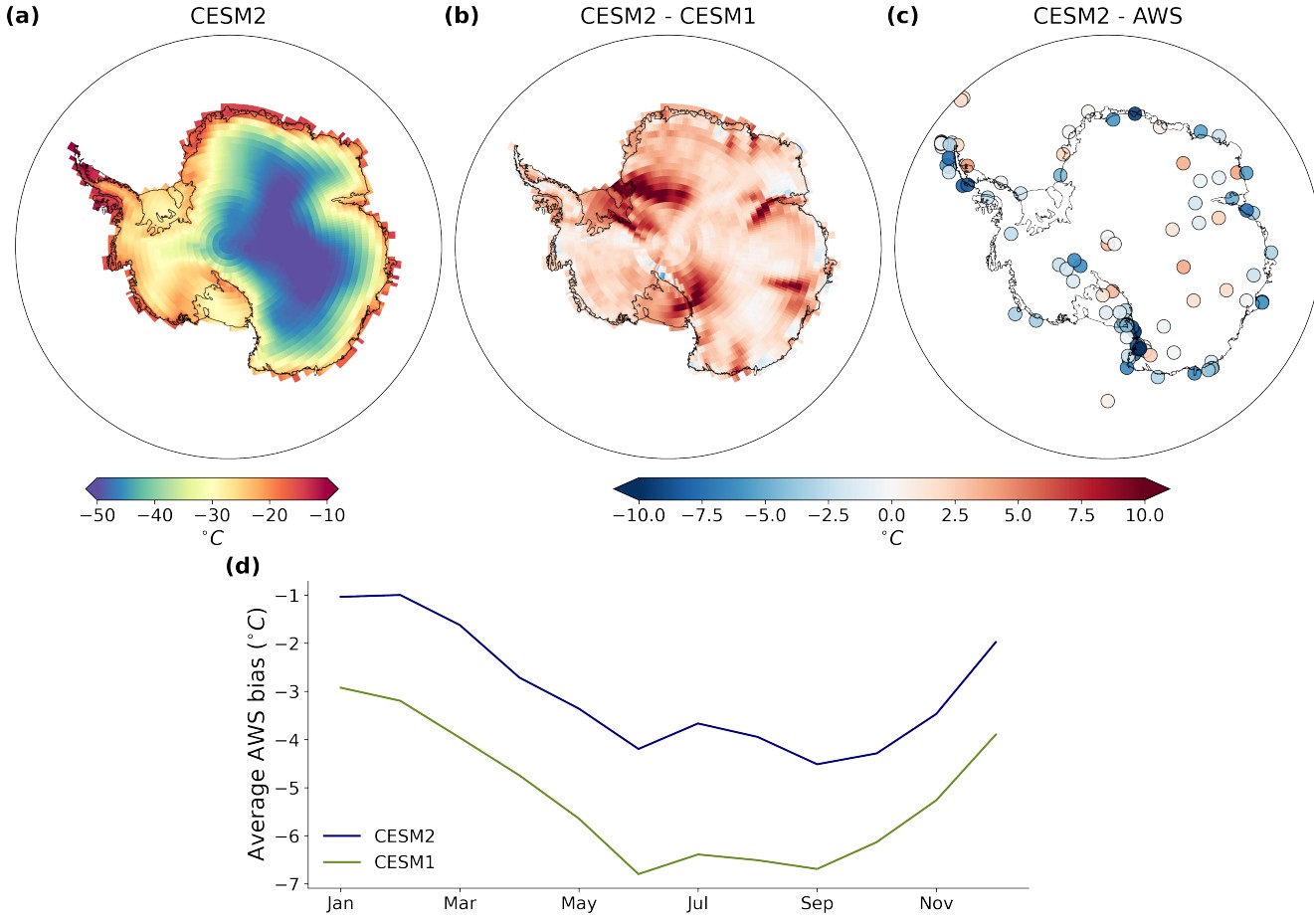

**Figure 1.** Comparison of CESM2 (1979-2015) AIS 2 m air temperature with CESM1 (1979-2005) and observations. (a) Average annual 2 m air temperature across the AIS from CESM2. (b) CESM2 - CESM1 modeled average annual 2 m temperature across the AIS. (c) Bias between CESM2 modeled 2 m air temperature and observations at 116 AWS locations. (d) Difference in monthly average 2 m air temperature between models (CESM2, CESM1) and AWS observations.

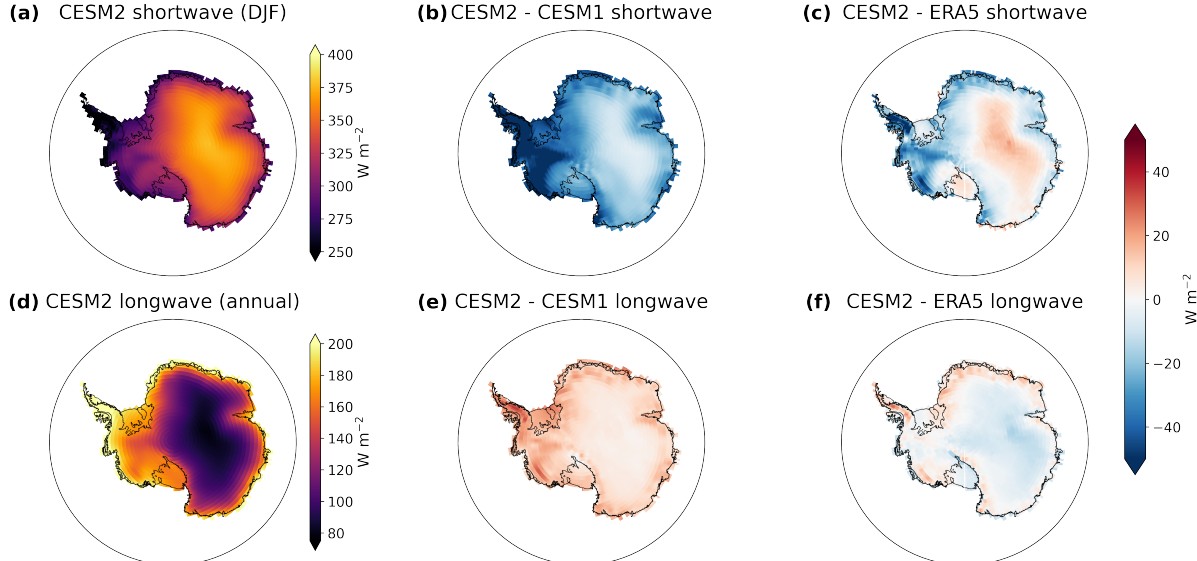

**Figure 2.** Comparison of incoming radiation components between CESM2 (1979-2015), CESM1 (1979-2005) and ERA5 (1979-2015). (a) CESM2 average austral summer incoming shortwave radiation. (b) CESM2 - CESM1 average austral summer incoming shortwave radiation. (c) CESM - ERA5 average austral summer incoming shortwave radiation. (d) CESM2 average annual incoming longwave radiation. (e) CESM2 - CESM1 average annual incoming longwave radiation. (f) CESM2 - ERA5 average annual incoming longwave radiation.

summer months whereas incoming longwave radiation impacts the surface energy balance year round, and (2) the high albedo
150  of snow reflects much of the incoming shortwave radiation back to space regardless. This phenomenon is evident in the model
as an increase in longwave radiation and a decrease in shortwave radiation, overall leads to an increase in net radiation and a
consequent increase in 2 m air temperature across the AIS (Fig. 1b), indicating that the longwave effect of clouds is dominant
in CESM2.

Compared with ERA5 (Fig. 2c,f), CESM2 has a spatially-averaged -7.3 W m$^2$ bias in incoming shortwave radiation (an
155  improvement from the +20.8 W m$^2$ CESM1 bias) and a -1.8 W m$^2$ bias in incoming longwave radiation (improved from
-12.2 W m$^2$ in CESM1). ERA5 suggests that CESM2 incoming shortwave radiation is negatively biased at the AIS coast
and positively biased in the interior (Fig. 2c), a spatial pattern that is consistent with CESM2 near-surface temperature biases
whereby modeled temperatures are largely too cold along the coast and too warm in the interior (Fig. 1c).

### 3.1.1 Historical temperature trends

160  Historical AIS near-surface temperature trends from CESM2 are in clear disagreement with those from ERA5. In ERA5, near-
surface temperatures have warmed significantly ($p < 0.05$) in the austral fall (MAM) over the western Antarctic Peninsula
($\sim$70 °W) and coastal Dronning Maud Land ($\sim$20 °W − 45 °E, DML), in the austral winter (JJA) over coastal DML, in the
austral spring (SON) over much of East Antarctica and the Ross ice shelf ($\sim$ 150 °W − 160 °E) and in the austral summer

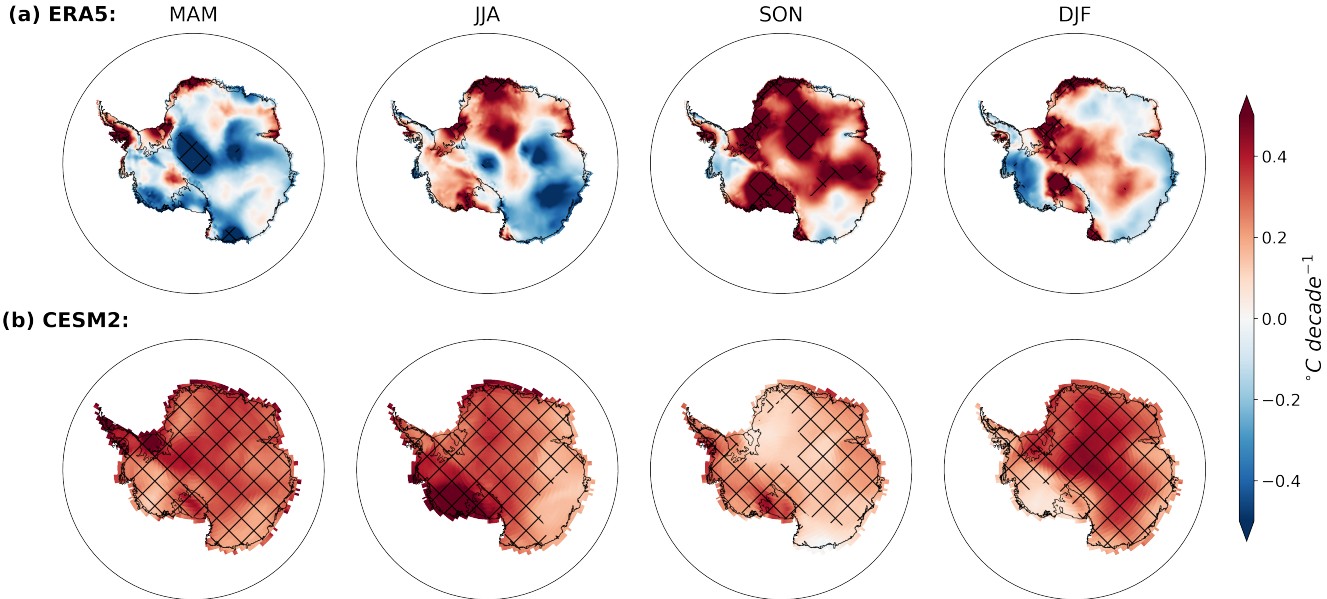

**Figure 3.** 1979 - 2015 seasonal temperature trends from (a) ERA5 and (b) CESM2. Cross-hatched areas represented regions where this trend is significant (p < 0.05).

(DJF) over the eastern edge of the Transantarctic mountains and coastal DML (Fig. 3a). Additionally, ERA5 near-surface temperatures have cooled significantly in MAM over small areas of East Antarctica. In contrast, CESM2 suggests significant near-surface warming across nearly the entire AIS in every season (Fig. 3b). While the austral fall (SON) has the smallest increasing temperature trend ($+0.18\,^{\circ}\mathrm{C}\ \mathrm{decade}^{-1}$) in CESM2, this season sees the largest warming trend ($+0.35\,^{\circ}\mathrm{C}\ \mathrm{decade}^{-1}$) in ERA5. In MAM, JJA, and DJF, ERA5 AIS temperature trends are -0.12, +0.03, and $+0.09\,^{\circ}\mathrm{C}\ \mathrm{decade}^{-1}$, respectively, while CESM2 AIS temperature trends for these same seasons are +0.31, +0.30, and $+0.28\,^{\circ}\mathrm{C}\ \mathrm{decade}^{-1}$.

## 3.2 Near-surface wind speed

Near-surface (10 m) wind speed on the AIS is greatest in the escarpment areas in East Antarctica, where steep slopes lead to more intense katabatic winds, a spatial signal that is well represented in the CESM2 annually averaged near-surface wind speed (Fig. 4a). Compared with CESM1, the spatially averaged annual AIS near-surface wind speed is $2.15 \pm 0.07\ \mathrm{m\,s^{-1}}$ higher in CESM2 (Fig. 4b). The largest wind speed increase between model versions occurs during the austral winter and spring (Fig. 4d), when wind speeds are typically the highest across the ice sheet. The overall wind speed increase in CESM2 leads to a better agreement with AWS observations (4c,d). In CESM2, the average annual near-surface wind speed bias between the model and observations at 96 different AWS locations is $+0.35\ \mathrm{m\,s^{-1}}$ (+5.0% relative bias), an improvement from an average bias of $-1.59\ \mathrm{m\,s^{-1}}$ in CESM1 (-22.6% relative bias). The CESM2 wind speed bias is consistently small ($<0.5\ \mathrm{m\,s^{-1}}$) throughout the year (Fig. 4d), indicating that CESM2 accurately portrays wind speed seasonality.

An improvement in wind speed from CESM1 to CESM2 also has implications for turbulent heat fluxes. The average annual latent heat flux across the AIS from 1979 - 2015 in CESM2 is $-1.6 \pm 0.1$ W m$^{-2}$, with positive values indicating a downward flux of energy (Fig. A3a). Spatially averaged, the latent heat flux from CESM2 is $1.1 \pm 0.1$ W m$^{-2}$ less than the latent heat flux from CESM1 (Fig. A3b), and is improved when compared with ERA5 ($-0.2$ W m$^{-2}$ bias for CESM2 and $+0.9$ W m$^{-2}$ bias for CESM1, Fig. A3c). The average annual AIS sensible heat flux in CESM2 is $23.3 \pm 0.3$ W m$^{-2}$ (Fig. A9d), $4.0 \pm 0.4$ W m$^{-2}$ greater than the sensible heat flux from CESM1 (Fig. A3e). Spatially averaged sensible heat flux in CESM2 is also improved from CESM1 when compared to ERA5, with average biases of $+0.1$ and $-3.8$ W m$^{-2}$ from CESM2 and CESM1, respectively (Fig. A3f). The spatial changes in sensible heat flux between model versions has further implications for near-surface air temperature. Where wind speed increases are minimal (e.g. edge of Filchner ice shelf, inland Amery ice shelf), more sensible heat is directed into the ice sheet, corresponding with relatively larger increases in temperature at these locations between the model versions.

### 3.3 Surface melt

#### 3.3.1 Comparison with QSCAT satellite observations

The average annual surface melt in CESM2 between 1979 and 2015 is $176.7 \pm 37.1$ Gt yr$^{-1}$ (Fig. 5b). While this is a substantial improvement from the annual CESM1 surface melt ($299.0 \pm 49.9$ Gt yr$^{-1}$, Fig. 5a), it is still 72.3 Gt yr$^{-1}$ greater than the average annual surface melt derived from the QSCAT satellite (104.3 Gt yr$^{-1}$, Fig. 5c). Total AIS surface melt from CESM2 is $69 \pm 35\%$ greater than observations, while AIS surface melt from CESM1 is $186 \pm 48\%$ greater than observations.

#### 3.3.2 Spatial melt patterns

In addition to showing a reduced bias in AIS annual surface melt magnitude, CESM2 is also much improved from CESM1 in representing spatial patterns of surface melt (Fig. 5). From QSCAT satellite-derived observations of surface melt, the Antarctic Peninsula (AP), West Antarctica (the West Antarctic Ice Sheet not including the AP, henceforth referred to as WAIS), and the East Antarctic Ice Sheet (EAIS) have 47.6, 13.2, 43.5 Gt yr$^{-1}$ of surface melt, respectively. CESM1 annual surface melt over the AP and WAIS is 25.0 and 5.2 Gt yr$^{-1}$ (47% and 60% less than observations, respectively), while annual surface melt from EAIS is 268.6 Gt yr$^{-1}$ (517 % larger than observations). Meanwhile, annual CESM2 surface melt from the AP, WAIS, and EAIS is 77.0 (62% larger than observations), 38.6 (193% larger than observations), 61.1 Gt yr$^{-1}$ (40% larger than observations), respectively. While EAIS surface melt is much more realistic in CESM2 than in CESM1, there has been a substantial increase in WAIS surface melt between the two model versions, which can be attributed too much melt on the Ronne-Filchner and Ross ice shelves.

Additionally, CESM2 shows a much more realistic distribution of surface melt over ice shelves vs. the grounded ice sheet. Both QSCAT observations and CESM2 indicate that the majority of surface melt occurs on ice shelves, with 72.2 Gt yr$^{-1}$ ice shelf melt from QSCAT and 124.1 Gt yr$^{-1}$ from CESM2 (72% larger than observations). By contrast, in CESM1 most surface melt occurs on the grounded ice sheet. Ice shelf melt from CESM1 is 65.6 Gt yr$^{-1}$ (9% less than observations) while grounded

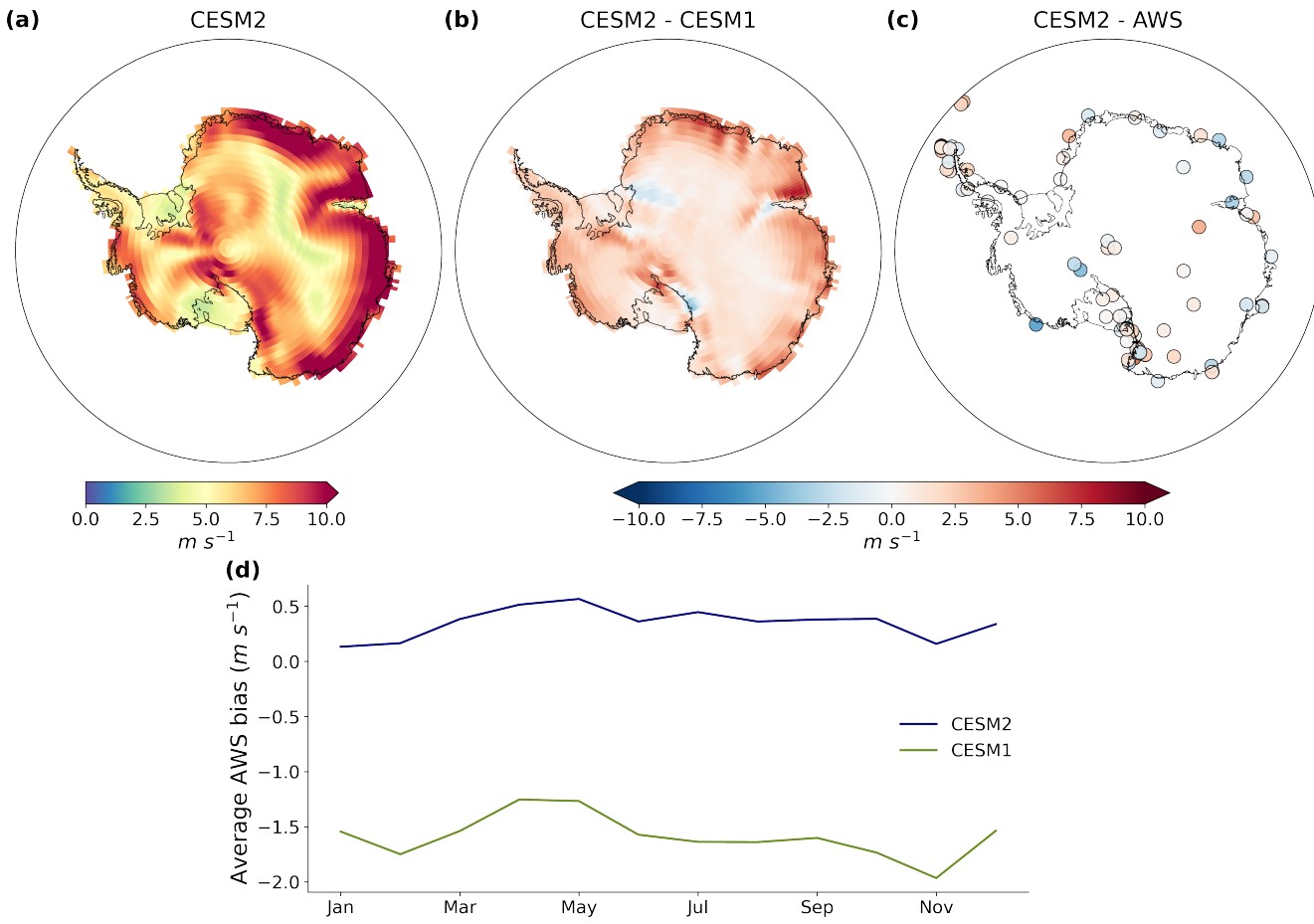

**Figure 4.** Comparison of CESM2 (1979-2015) AIS 10 m wind speed with CESM1 (1979-2005) and observations. (a) Average annual 10 m wind speed across the AIS from CESM2. (b) CESM2 - CESM1 modeled average annual 10 m wind speed across the AIS. (c) Bias between CESM2 modeled 10 m wind speed and observations at 98 AWS locations. (e) Difference in monthly average 10 m wind speed between models (CESM2, CESM1) and AWS observations.

ice sheet melt is 233.2 Gt yr$^{-1}$, 626% larger than QSCAT observations suggest. CESM2 has a substantially improved ratio of ice shelf to grounded ice sheet melt; however, CESM2 surface melt typically does not extend as far into the interior ice sheet as observations suggest (Fig. 5d). This lack of modeled interior melt is relatively small compared to the melt that occurs closer to the coast and is likely due to coarse model resolution.

### 3.3.3 Historical melt trends

Historical (1979-2015) surface melt in CESM2 has increased across much of the AIS (Fig. 5e), a trend that is absent from both regional climate model estimates of melt and microwave satellite observations of melt duration and area (Kuipers Munneke et al., 2012). In CESM2, a trend dipole exists in WAIS, whereby surface melt has increased over the Ronne-Filchner, Pine Island, and Thwaites ice shelves, and decreased inland and over the Ross ice shelf (Fig. 5e). A similar pattern in austral summer (DJF) near-surface temperature trends exists (Fig. 3b), with near-surface temperature increasing relatively less over inland WAIS and the Ross ice shelf. The surface melt and near-surface temperature trend dipole is caused by an increasing Southern Annular Mode (SAM) which is due, in part, to intensifying Antarctic ozone depletion (Lenaerts et al., 2018). The increasing DJF SAM is evident in CESM2 by increasing DJF meridional sea level pressure gradient, whereby sea level pressure is decreasing close to the AIS and increasing at lower latitudes near 50°S (Fig. A4a), and in decreasing DJF geopotential height surrounding the AIS (Fig. A4b) and increasing DJF westerly winds around 60°S (Fig. A4c).

### 3.4 Surface mass balance

### 3.4.1 Comparison of the mean surface mass balance with other products

In CESM2, the annual average grounded surface mass balance (SMB) between 1979 and 2015 is 2269 $\pm$ 100 Gt yr$^{-1}$ (Fig. 6a), significantly (p < 0.05) greater than the average annual grounded SMB from CESM1 (1790 $\pm$ 85 Gt yr$^{-1}$), ERA5 (1960 $\pm$ 106 Gt yr$^{-1}$), RACMO2.3 (1997 $\pm$ 93 Gt yr$^{-1}$), MAR (2150 $\pm$ 96 Gt yr$^{-1}$), and the MT2019 reconstruction (1788 $\pm$ 293 Gt yr$^{-1}$). We also compared CESM2 (from CMIP6) with the 100-member CESM2-LENS and found that both models produce similar estimates of AIS SMB (Fig. 6a).

Over ice shelves, CESM2 has an average SMB of 559 $\pm$ 27 Gt yr$^{-1}$ between 1979 and 2015, significantly greater (p < 0.05) than the average ice shelf SMB from CESM1 (520 $\pm$ 26 Gt yr$^{-1}$), ERA5 (506 $\pm$ 26 Gt yr$^{-1}$), RACMO2.3 (523 $\pm$ 24 Gt yr$^{-1}$), and MAR (459 $\pm$ 23 Gt yr$^{-1}$). The MT2019 reconstruction only covers the grounded ice sheet and thus ice shelf SMB cannot be calculated from this product.

For the full ice sheet, accumulation from both solid and liquid precipitation accounts for 91.7% of the total SMB signal in CESM2, with ablation terms accounting for 8.3% of the signal (6.5% from sublimation/evaporation and 1.8% from runoff). This breakdown is comparable to that from ERA5, where 92.1% of the total SMB signal comes from precipitation, 6.9% from sublimation/evaporation, and 1.0% from runoff. In comparison, only 2.0% of the total SMB signal from CESM1 comes from sublimation/evaporation (with 96.6% from precipitation and 1.4% from runoff). This increase in the sublimation/evaporation contribution to the SMB signal from CESM1 to CESM2 is likely due to the increase in near-surface wind speed (discussed in

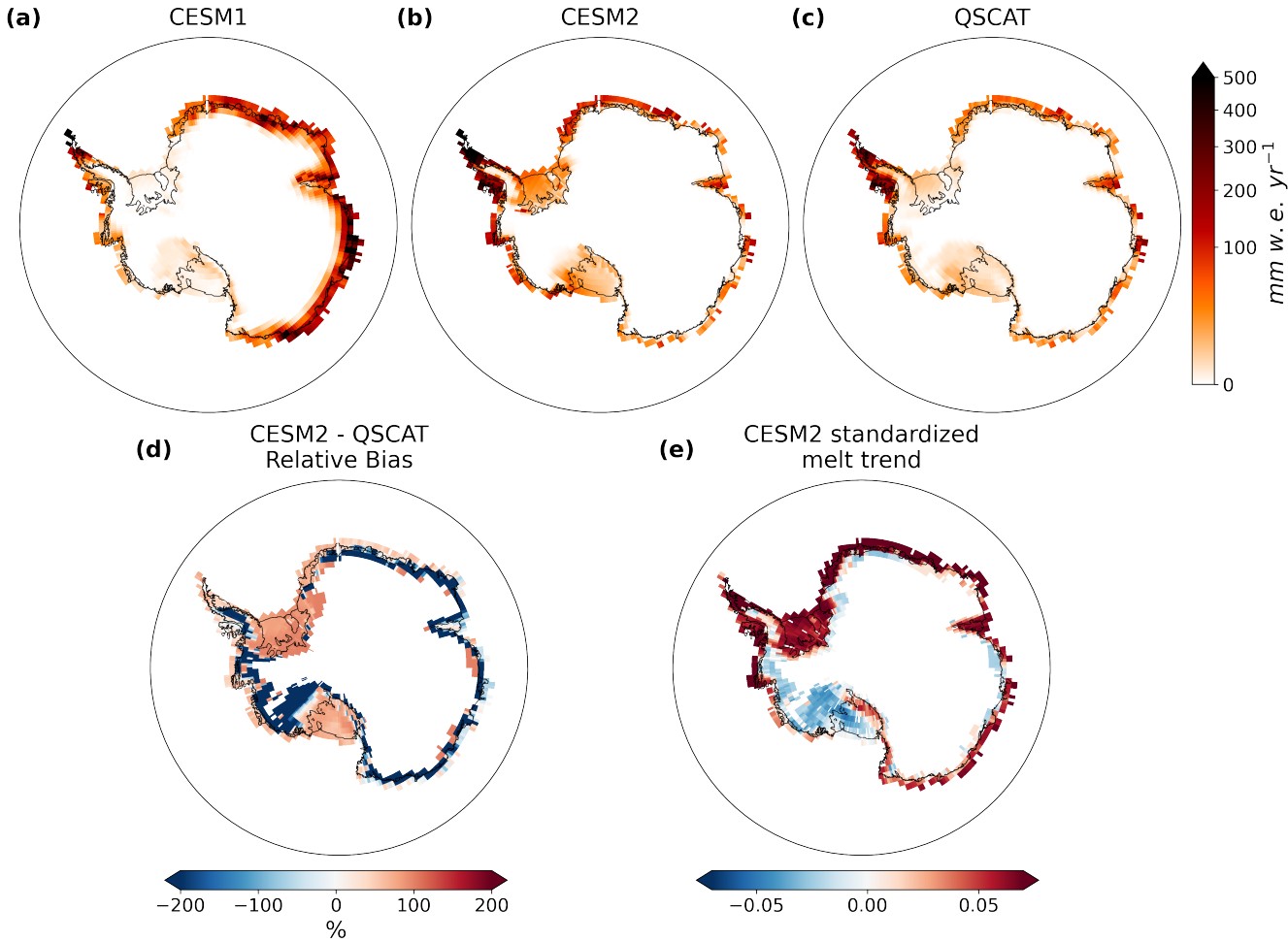

**Figure 5.** Melt from CESM1, CESM2 and observations. (a) 1979-2005 average annual surface melt from CESM1. (b) 1979-2015 average annual surface melt from CESM2. (c) 1999-2009 average annual surface melt derived from the QSCAT satellite (Trusel et al., 2013). (d) CESM2 - QSCAT relative bias. (e) 1979-2015 standardized CESM2 historical melt trend.

Section 3.2, Fig. 4b) which drives a corresponding decrease in positive-downward latent heat flux between the model versions
(Fig. A3b).

### 3.4.2 Spatial SMB patterns

Spatially, SMB increases from the dry, high elevation interior of the AIS to the coastal regions and ice shelves that receive more
annual precipitation (Fig. 6b,c). Spatially averaged annual SMB in CESM2 is the largest in the AP at 572 mm water equivalent
(w.e.) per year, followed by WAIS (303 mm w.e. $yr^{-1}$). EAIS, being drier than both WAIS and the AP, has the lowest modeled
average SMB (105 mm w.e. $yr^{-1}$). DML and Enderby Land (45 °E − 60 °E, EL) are the primary regions responsible for
the greater SMB in CESM2 compared to the MT2019 reconstruction (Fig. 6d). Combined, QML and EL drainage basins 4-8
(Zwally et al., 2012, Fig. A5) have 195 Gt $yr^{-1}$ (+34%) higher SMB in CESM2 than in the MT2019 reconstruction (Fig. 6d).

### 3.4.3 Historical SMB trends

A major difference in SMB between CESM2 and the MT2019 reconstruction, reanalysis, and regional climate models is that
there is a positive SMB trend in CESM2 (as well as in CESM1) that is absent in any other products used in this study. Prior to
1971, CESM2 has a significantly positive (p < 0.05) AIS SMB trend of 0.53 Gt $yr^{-2}$. After 1971, the model has a significantly
positive SMB trend of 4.69 Gt $yr^{-2}$. We consider 1971 as a 'breakpoint year' because the change in SMB trend between
preceding and subsequent 30 year time periods is the greatest in the 1850-2015 year period (Fig. A6).

The positive SMB trend in CESM2 is driven by increasing precipitation, particularly in DJF and in QML, East Antarctica
(Fig. 7b). Along the coast of QML, DJF precipitation has increased significantly (p < 0.05), upwards of 1 mm w.e $yr^{-2}$
since 1979. In WAIS, a precipitation trend dipole (similar to the melt and temperature trend dipole discussed in Section 3.3.3)
appears in CESM2 in MAM, and even more prominently in DJF, whereby precipitation has decreased over the Ross ice shelf
and surroundings and increased over eastern WAIS, including the Amundsen (∼105 °W) and Bellinghausen (∼80 °W) sea
regions, and the Ronne-Filchner ice shelf (Fig. 7b). DJF precipitation has decreased insignificantly in WAIS basins 18 and 19
(Zwally et al., 2012, Fig. A5) by 0.96 Gt $yr^{-2}$ from 1979 to 2015 (Fig. A7). Meanwhile, neighboring basins 1, 22, and 23 have
seen a significant (p < 0.05) 2.52 Gt $yr^{-2}$ increase in DJF precipitation during this same period (Fig. A7). In comparison with
ERA5, the precipitation dipole appears stronger in ERA5 in MAM and is non-existent in ERA5 in DJF (Fig. 7a).

AIS historical precipitation trends in CESM2 appear to be largely driven by the increasing SAM and intensifying Antarctic
ozone depletion, with spatial patterns similar to that shown in Lenaerts et al. (2018). Strong increasing DJF precipitation trends
(as a result of ozone depletion) are found over the inland eastern WAIS, western coastal DML (∼30 °W − 0 °W), and the
Amery drainage basin (∼60 °E − 70 °E), while significant ozone-depletion-forced decreasing DJF precipitation trends exist
in western WAIS and over the Transantarctic mountains (Lenaerts et al., 2018). Further, decreasing geopotential height within
CESM2 (Fig. A4b) has likely led to increasing precipitation across much of the AIS.

Differences in historical precipitation trend between ERA5 and CESM2 exist across much of the AIS, but particularly in
Wilkes Land and Princess Elizabeth Land (∼75 °E − 136 °E), with precipitation largely decreasing in ERA5 but increasing
in CESM. Additionally, over the eastern AP (∼63 °W) in DJF, precipitation decreases strongly in ERA5 but remains roughly

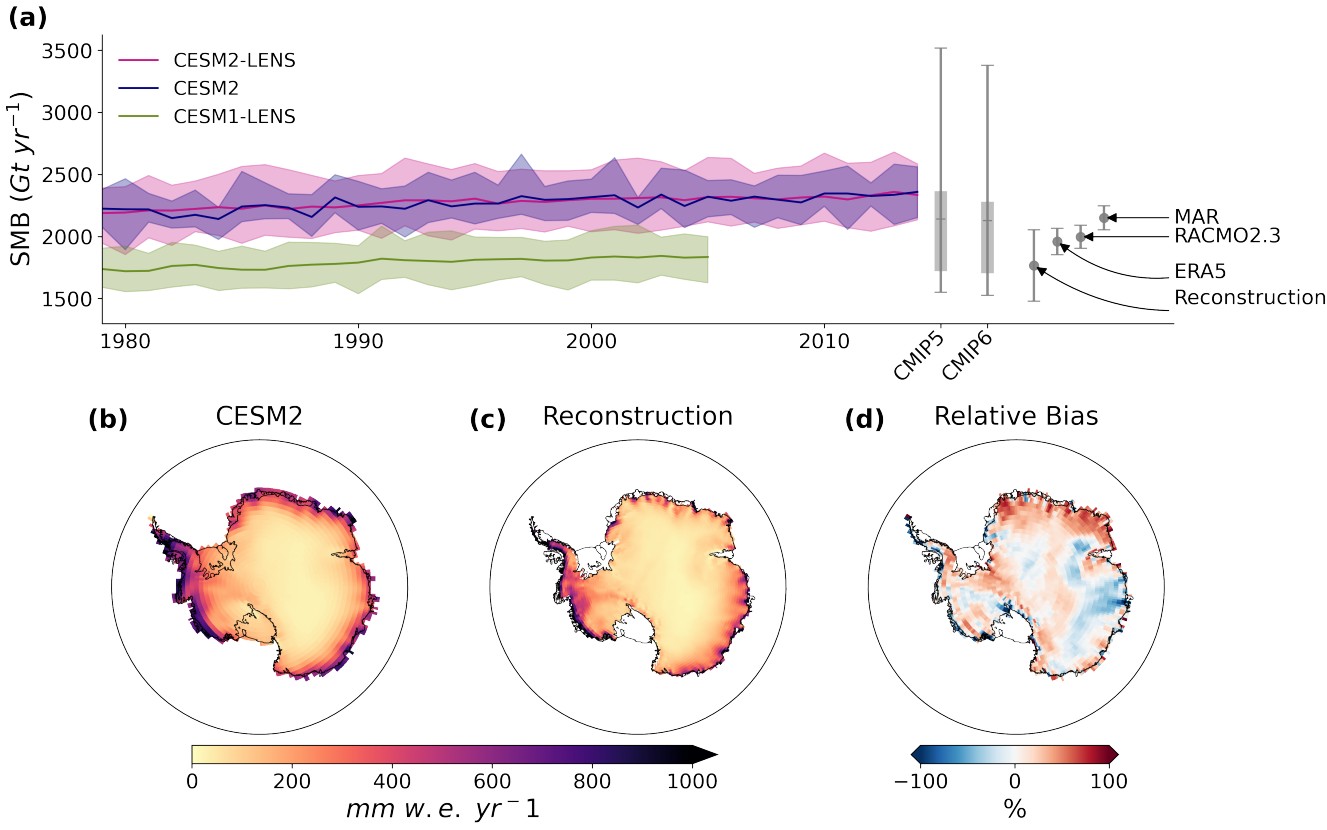

**Figure 6.** Modeled AIS SMB. (a) 1979-2015 time series of annual grounded AIS SMB from CESM2-LENS, CESM2, and CESM1-LENS with ensemble mean plotted with the solid line and ensemble spread shaded. The average annual SMB spread for all CMIP5 and CMIP6 models is shown on the right with grey box and whiskers plots. Also shown is the average annual SMB from the MT2019 reconstruction with error bars representing reconstruction error and the average annual SMB from MAR, RACMO2.3, and ERA5 with error bars representing ± 1 standard deviation. (b) 1979-2015 annual AIS SMB from CESM2. (c) 1979-2000 annual AIS SMB from the MT2019 reconstruction. (d) Relative bias between CESM2 and MT2019 reconstruction SMB.

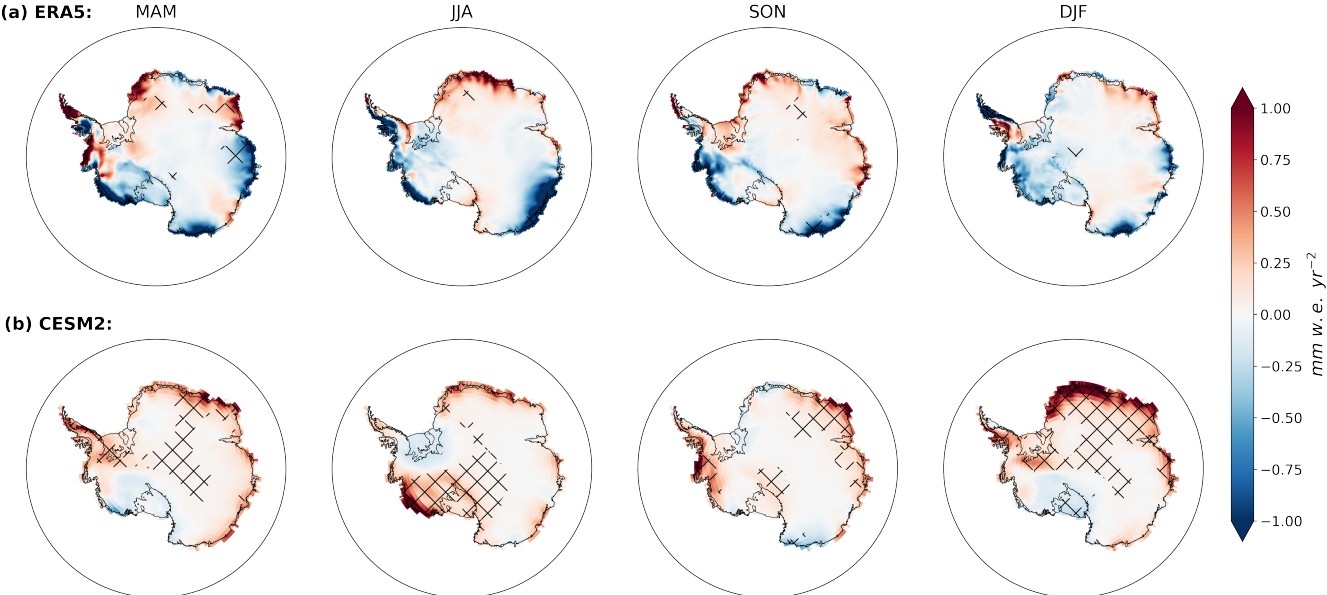

**Figure 7.** 1979-2015 trend in seasonal precipitation from (a) ERA5, and (b) CESM2. Cross-hatched areas represent regions where this trend is significant (p < 0.05).

constant in CESM2. The difference in precipitation trend over the AP may be due to unresolved topography in the larger CESM2 grid cells.

### 3.5 Future model trends

280 Keeping historical CESM2 model biases in AIS surface climate means and trends in mind, here we investigate future sim-
ulations of AIS SMB under three different climate change scenarios (Meehl et al., 2020). According to CESM2, increasing
atmospheric temperatures throughout the 21st century are expected to increase precipitation across the AIS, which will corre-
spond with future increases in AIS SMB. Forced with the high-emission scenario (SSP5-8.5), near-surface air temperature over
the full ice sheet increases by 6.7 °C from the final ten years of the historical simulation (2005-2015) to the final ten years of

285 the scenario (2090-2100), while annual SMB increases by 637 Gt yr$^{-1}$. In the middle- and low-emission scenarios (SSP3-7.0
and SSP1-2.6, respectively), the near surface air temperature increases by 4.9 °C and 1.8 °C and the annual SMB increases by
569 and 289 Gt yr$^{-1}$. The 21st century change in SMB with respect to change in temperature ($\frac{\Delta SMB}{\Delta T}$) over the full ice sheet
is +94 Gt yr$^{-2}$ °C$^{-1}$ from SSP5-8.5, +116 Gt yr$^{-2}$ °C$^{-1}$ from SSP3-7.0, and +159 Gt yr$^{-2}$ °C$^{-1}$ from SSP1-2.6.

A diverging future SMB trend on ice shelves and the grounded ice sheet, of which CESM2 agrees with previous studies (Kit-

290 tel et al., 2021), is responsible for the varying $\frac{\Delta SMB}{\Delta T}$ between different emission scenarios. On the grounded ice sheet, SMB
increases approximately linearly with increasing temperatures (Fig. 8d, A8) at rates of +123, +130, and +147 Gt yr$^{-2}$ °C$^{-1}$
for the SSP5-8.5, SSP3-7.0, and SSP1-2.6 scenarios respectively. In contrast, on ice shelves, SMB begins to decrease with

increasing temperatures around the year 2060 in the SSP5-8.5 and SSP3-7.0 scenarios (Fig. 8e, A8). In SSP5-8.5 and SSP3-7.0 ice shelf $\frac{\Delta SMB}{\Delta T}$ is -50 and -30 Gt yr$^{-2}$ °C$^{-1}$, respectively, while the ice shelf $\frac{\Delta SMB}{\Delta T}$ for SSP1-2.6 is +5 Gt yr$^{-2}$ °C$^{-1}$. As temperature increases, melt and rainfall increase non-linearly, depleting the pore space in the ice-shelf firn, and increasing runoff, which begins to dominate the SMB signal. Forced with SSP5-8.5, CESM2 indicates that approximately 40% of AIS liquid production (melt and rainfall) leaves the ice sheet as meltwater runoff by 2100, compared with only 10% at the beginning of the simulation (Fig. A9). On ice shelves specifically, more than 50% of the total meltwater produced at the surface runs off, indicating that runoff has surpassed refreezing by the end of the century. Increasing runoff on ice shelves can explain a more-than-linear decrease in ice shelf SMB (Fig. 8e, A8). Interestingly, this divergence in SMB trend on ice shelves is not projected to occur in the low-emission scenario, in which increasing snowfall appears to be sufficient in mitigating enhanced melt and preventing firn pore space depletion, thus limiting runoff in this scenario.

At the end of the historical simulation (2005-2015), solid precipitation contributes to 91.7% of the total grounded SMB signal in CESM2, while rainfall, evaporation/sublimation, and runoff contribute 0.7%, 6.1%, and 1.5% respectively (Fig. A10). By the end of the future period (2090-2100), the contribution of both rainfall and runoff to the modeled SMB signal increases slightly in all scenarios (3.1% and 7.1%, respectively in SSP5-8.5), with a corresponding decrease in the contribution of precipitation (83.1% in SSP5-8.5). Over ice shelves, we see a much greater change in the contribution of these different components to the total CESM2 SMB signal at the end of the future period (Fig. A10). From 2005 to 2015, snowfall accounts for 77.6% of the modeled ice shelf SMB signal, rainfall accounts for 5.4%, evaporation/sublimation accounts for 7.0%, and runoff accounts for 10.0%. By the end of the SSP5-8.5 scenario, snowfall accounts for less than half of the ice shelf SMB signal (41.8%), with rainfall, evaporation/sublimation, and runoff accounting for 14.8%, 3.9%, and 39.5%, respectively.

The SMB seasonal cycle also changes in future scenarios, becoming more amplified with increased warming (Fig. 9). For both ice shelves and the grounded ice sheet, increased JJA temperatures increase solid precipitation and therefore SMB, as melt and liquid precipitation remain confined to the austral summer season. Average JJA SMB increases by ∼79 Gt yr$^{-1}$ from the last 10 years of the historical simulation (2005-2015) to the last ten years of the future SSP5-8.5 simulation (2090-2100) over the grounded ice sheet, and increases by ∼35 Gt yr$^{-1}$ over ice shelves in the same scenario. In contrast, during DJF, atmospheric warming leads to decreased SMB as melt, and therefore runoff, increases. On ice shelves, we see increasingly negative DJF SMB in the three future scenarios. For example, from the last 10 years of the historical simulation to the last 10 years of the future SSP5-8.5 simulation, DJF ice shelf SMB decreases from ∼22 Gt yr$^{-1}$ to ∼-101 Gt yr$^{-1}$, further amplifying SMB seasonality.

## 4  Discussion and conclusions

In this paper we have analyzed the surface climate in different regions of Antarctica, including the Antarctic Peninsula (AP). However, since the AP consists of complex topography that is challenging to resolve with the CESM2 horizontal resolution, caution is warranted regarding the simulation of the AP climate in CESM2. To advance our understanding of the AP surface

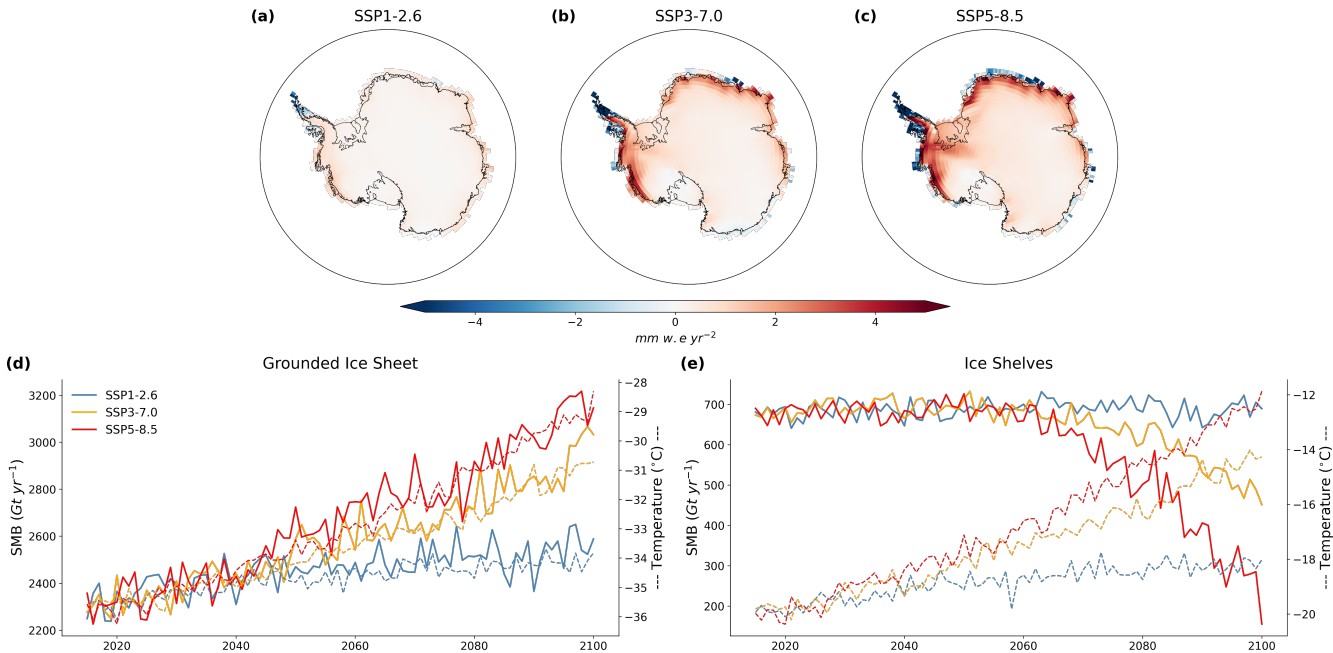

**Figure 8.** Future (2015-2100) SMB in CESM2. (a-c) SMB trend from low (SSP1-2.6), middle (SSP3-7.0) and high (SSP5-8.5) socioeconomic pathways. (d) Timeseries of annual grounded SMB (left axis, solid lines) and temperature (right axis, dashed lines) from different CESM2 SSPs. (e) Timeseries of annual SMB and temperature over ice shelves.

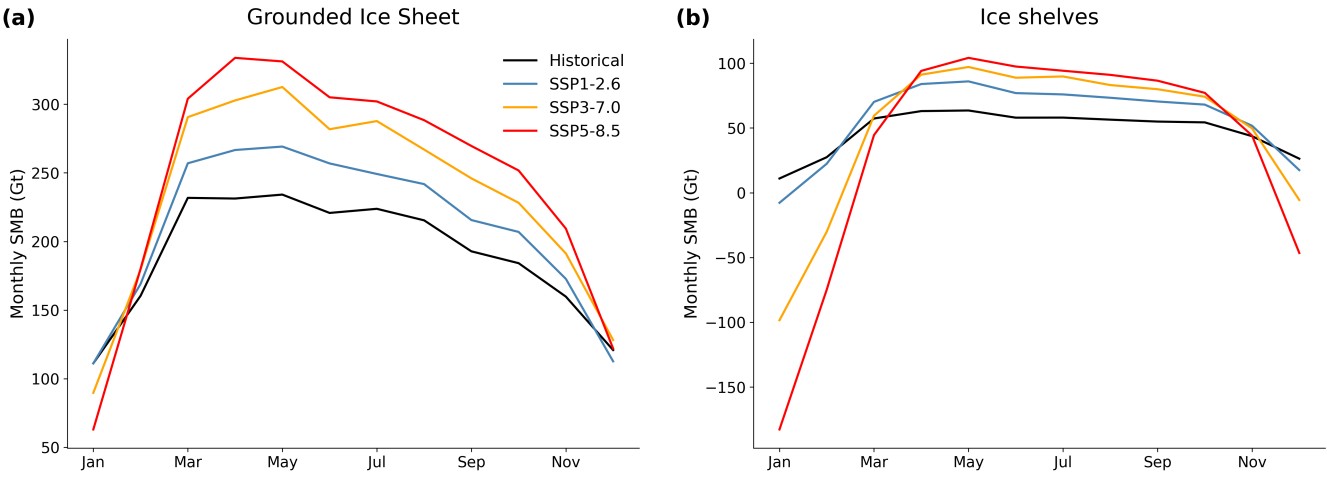

**Figure 9.** Seasonality of SMB in the last decade of historical (2005-2015) and future (2090-2100) SSP model output over the grounded ice sheet (a) and ice shelves (b).

climate, improved model resolution is necessary (van Lipzig et al., 2004; Van Wessem et al., 2016; Turton et al., 2017; Datta et al., 2018).

Overall, model updates between CESM1 and CESM2, particularly in cloud physics, snow model, and orographic drag representation, result in a lower CESM2 bias, compared to CESM1, with regards to mean-state near-surface temperature, wind speed, surface melt, and incoming radiation. One major improvement in CESM2 is a reduction in overall AIS surface melt
volume and a more realistic spatial distribution of melt compared with CESM1 (Fig. 5). We attribute this to improvements in the snow component of the land model (van Kampenhout et al., 2017). Although melt in CESM2 is much improved, total annual melt volume across the AIS is still substantially higher than observations, indicating that further improvements with the snow model or the atmospheric forcing of surface melt are necessary.

Another CESM2 improvement is that near-surface temperatures are closer to observations (Fig. 1). This improvement results
from CESM2 enhanced cloud liquid water due to upgraded cloud microphysical parameterizations in polar regions (Lenaerts et al., 2020). These model upgrades have also led to a relatively small decrease in incident shortwave radiation (Fig. 2b) and a larger increase in incident longwave radiation (Fig. 2d) across the AIS, resulting in net increased cloud radiative forcing, net surface warming, and more realistic near-surface temperatures.

However, changes in cloud microphysical parameterizations have simultaneously increased annual precipitation in CESM2,
resulting in annual precipitation that is too high and unrealistic when compared with observations. Average annual precipitation in CESM2 between 1979 and 2015 is $29 \pm 7.3\%$ higher than in CESM1, $15 \pm 6.8\%$ higher than in ERA5, and $13 \pm 6.3\%$ higher than in RACMO2.3 (compared with CESM1 which is $11 \pm 6.2\%$ lower than ERA5, and $13 \pm 5.6\%$ lower than RACMO2.3). Excessive precipitation results in an unrealistically high SMB and highlights an area of improvement for future model versions.

A second unrealistic behavior of CESM2 is the historical trend in precipitation, and therefore SMB, that cannot be recon-
ciled with observations. From 1971 to 2015, CESM2 SMB increased at a rate of $4.69$ Gt $\mathrm{yr}^{-1}$, a trend that is absent from other reanalysis, reconstruction, and regional climate modeling products used in this study. The unrealistic precipitation increase is likely due to the high climate sensitivity of CESM2 (Gettelman et al., 2019). Zhu et al. (2022) find that the CESM2 climate is very sensitive to treatments of cloud microphysical processes and that tuning these processes results in a modeled climate sensitivity that more realistically matches present-day observations. CESM2's high climate sensitivity likely implies
that modeled future precipitation and runoff trends are also overestimated, something that should be taken into consideration when discussing CESM2 AIS SMB under different future emissions scenarios.

In the context of the larger Southern Hemisphere (SH), Dalaiden et al. (2020) show that the CESM2 Antarctic moisture budget due to synoptic and large-scale atmospheric circulation is realistic compared to reanalysis (ERA-Interim). This indicates that too-high CESM2 mean-state precipitation may be attributed to cloud microphysics, not SH moisture budget. While CESM2
performs well regarding the mean-state SAM and the location of the SH jet, its representation of stationary waves and the speed of the SH jet have degraded from CESM1 (Simpson et al., 2020). Zonal circulation appears overall too strong in CESM2, which may enhance or reduce precipitation in various regions across the AIS. Analogous to the unrealistic precipitation trend in CESM2, there is also a decrease in CESM2 SH sea ice throughout the historical period that cannot be reconciled with observations (DuVivier et al., 2020; Raphael et al., 2020). The unobserved SH sea ice and AIS precipitation trends may arise

from similar factors (i.e. high CESM2 climate sensitivity); and/or, a decrease in sea ice may contribute to increasing AIS precipitation.

    In future emissions scenarios, we find an important divergence in CESM2-simulated SMB trend between ice shelves and the grounded ice sheet. While SMB over the grounded ice sheet continues to increase linearly with temperature in all future scenarios, ice-shelf SMB begins to decrease rapidly beginning in approximately 2060 due to a non-linear increase in surface

melt and runoff. Although we acknowledge the positive melt bias in CESM2 during the historical period which likely impacts the representation of melt and runoff in future scenarios, this is a phenomenon that has similarly been modeled with MAR (Gilbert and Kittel, 2021; Kittel et al., 2021). The rapid SMB decline on ice shelves is important because ice shelves buffer the inland flow of ice from the grounded ice sheet, mitigating its contribution to sea level rise, and with decreasing SMB, are vulnerable to collapse in a warming climate. While CESM2's firn model has improved substantially (van Kampenhout et al.,

2017), it still only allows for a ~20-30 meters deep firn column, which likely results in an underestimation of meltwater storage capacity in the firn across much of the AIS. In a future warming climate with non-linearly increasing meltwater production on Antarctic ice shelves, CESM2 may exaggerate runoff as a result of this shallow firn column, highlighting the need for continued development of the snow model to better understand future SMB changes.

    Recently, there has been some work done to couple ice sheet models and ESMs (Siahaan et al., 2021). However, even in the

latest iteration of estimating future AIS contribution to sea level rise, Antarctic ice sheet models are largely simulated as a stand-alone, meaning they require climate forcing (Seroussi et al., 2020). CMIP6 ESMs such as CESM2 will be more extensively used as this forcing for ice sheet models (Payne et al., 2021). Further, CESM2 does not have an interactive AIS; however, this is a high priority for the CESM community as it prepares for the next version, CESM3. With this goal in mind, the model will need realistic climate forcing. Here we show that CESM2 sees an improvement in mean-state near-surface temperature and

wind speed, melt, and incoming radiation components compared with CESM1 due to an improved snow model and upgraded cloud microphysical parameterizations. However, CESM2 has a corresponding downgrade in annual precipitation amount, with exaggerated precipitation compared to other reanalysis, reconstruction, and regional climate modeling products. Similarly, a significantly positive precipitation trend between 1971 and 2015 does not match observations and highlights the high climate sensitivity of CESM2. These two factors should be future areas of focus when preparing for CESM3.

**Appendix A: Historical model results**

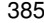

**Figure A1.** 1979-2015 seasonal trend in near-surface temperature from ERA5 (a), RACMO2.3 (b), and MAR (c). Cross-hatched areas represent regions where this trend is significant (p < 0.05).

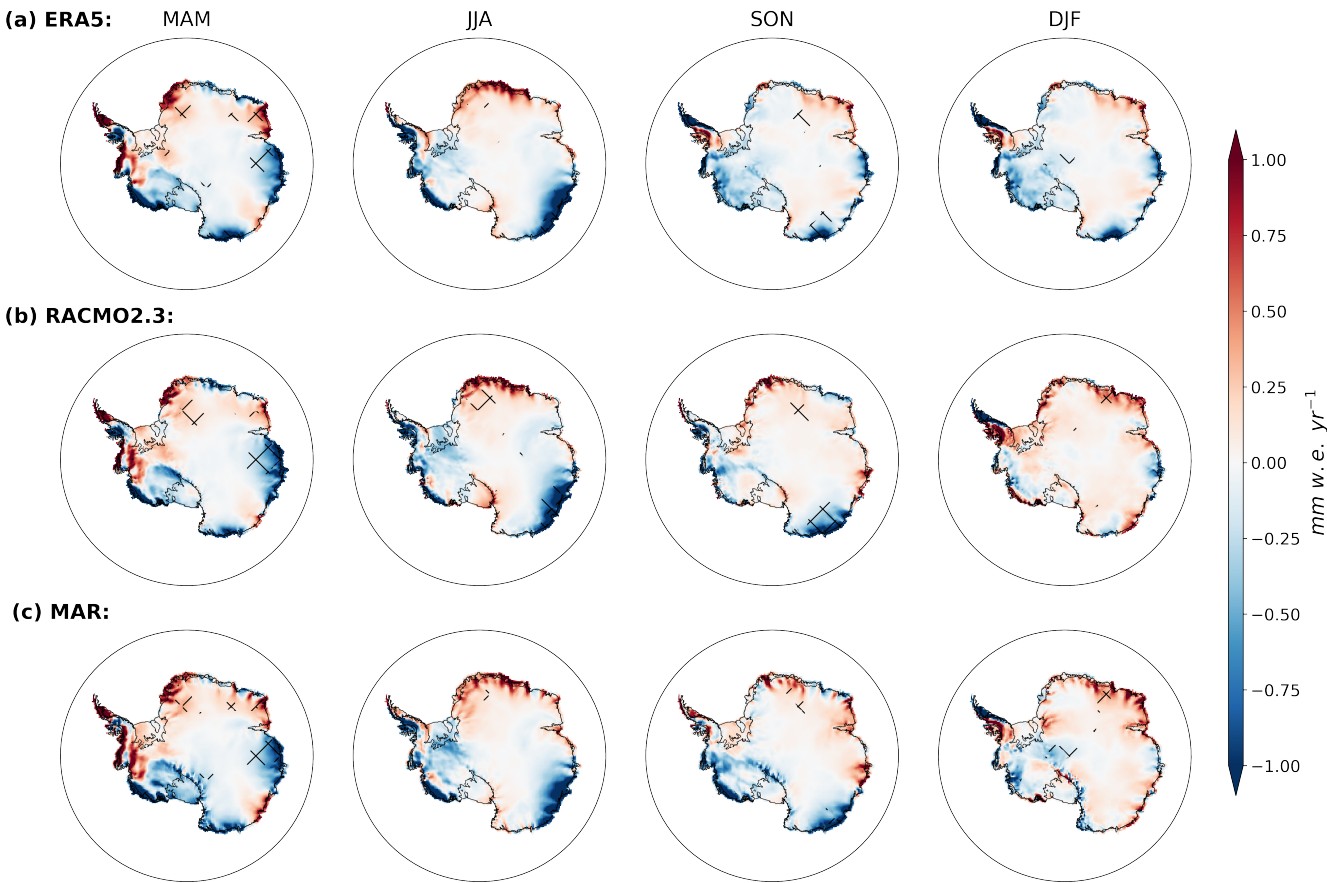

**Figure A2.** 1979-2015 seasonal trend in precipitation from ERA5 (a), RACMO2.3 (b), and MAR (c). Cross-hatched areas represent regions where this trend is significant (p < 0.05).

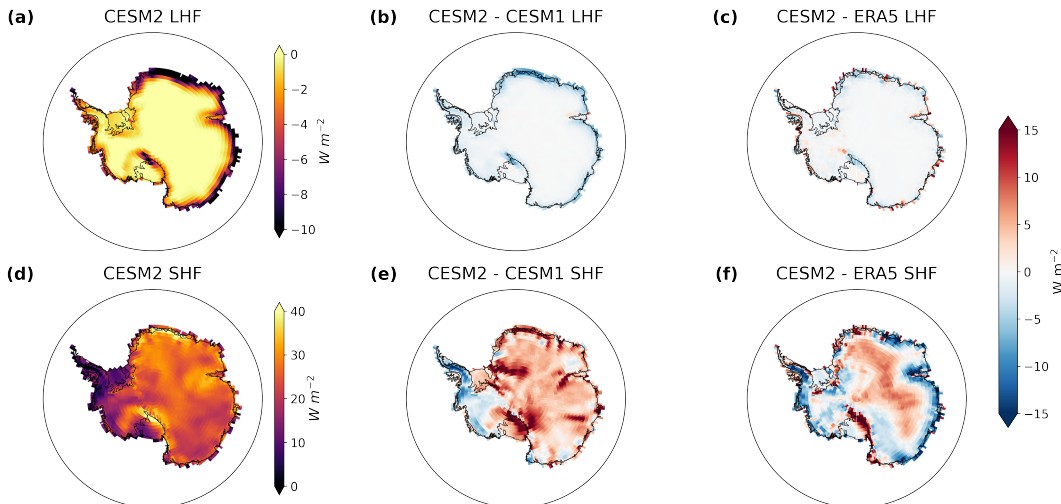

**Figure A3.** Comparison of turbulent fluxes between CESM2 (1979-2015), CESM1 (1979-2005) and ERA5 (1979-2015). (a) CESM2 average annual latent heat flux (LHF). (b) CESM2 - CESM1 average annual LHF. (c) CESM - ERA5 average annual LHF. (d) CESM2 average annual sensible heat flux (SHF) (e) CESM2 - CESM1 average annual SHF. (f) CESM2 - ERA5 average annual SHF. Positive values indicate a downward net energy flux (into the ice sheet).

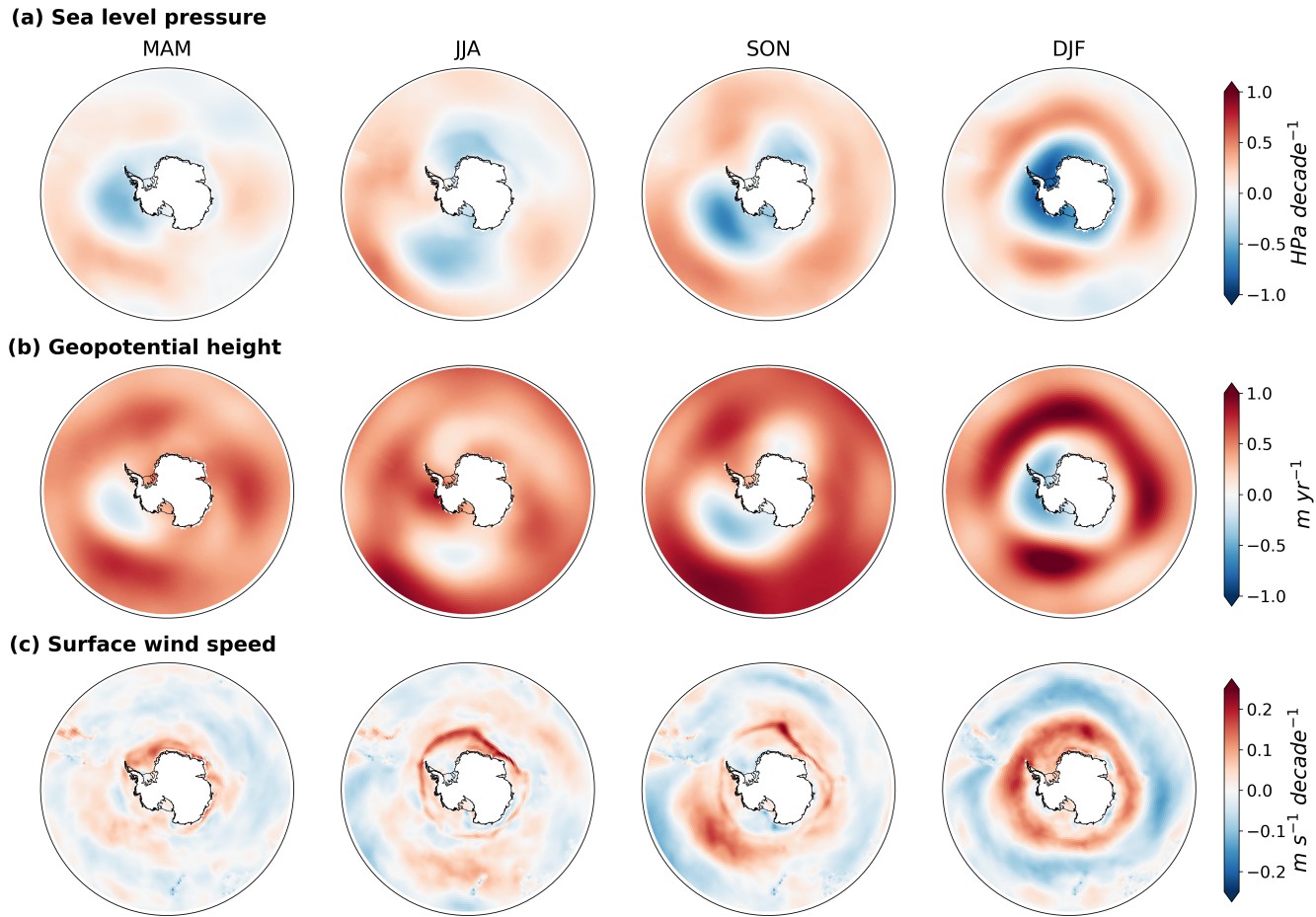

**Figure A4.** 1979-2015 CESM2 seasonal trends in (a) sea level pressure around the AIS, (b) 500 hPa geopotential height, and (c) surface wind speed over the Southern Ocean.

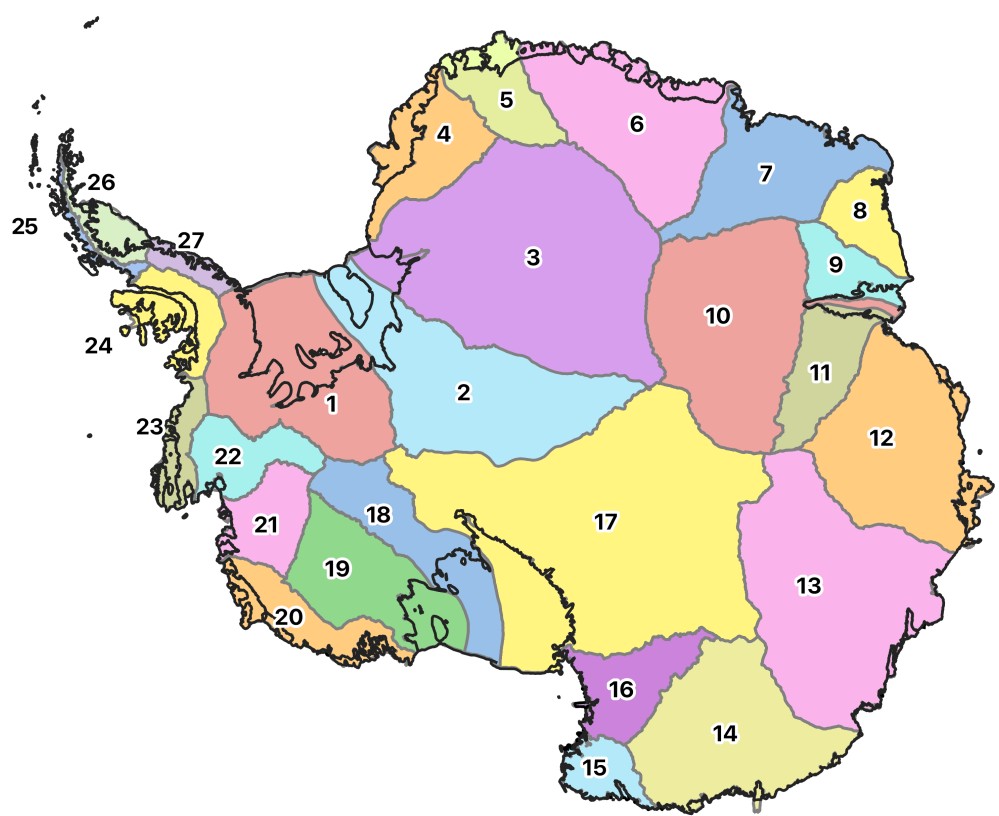

**Figure A5.** Labelled AIS drainage basins (Zwally et al., 2012).

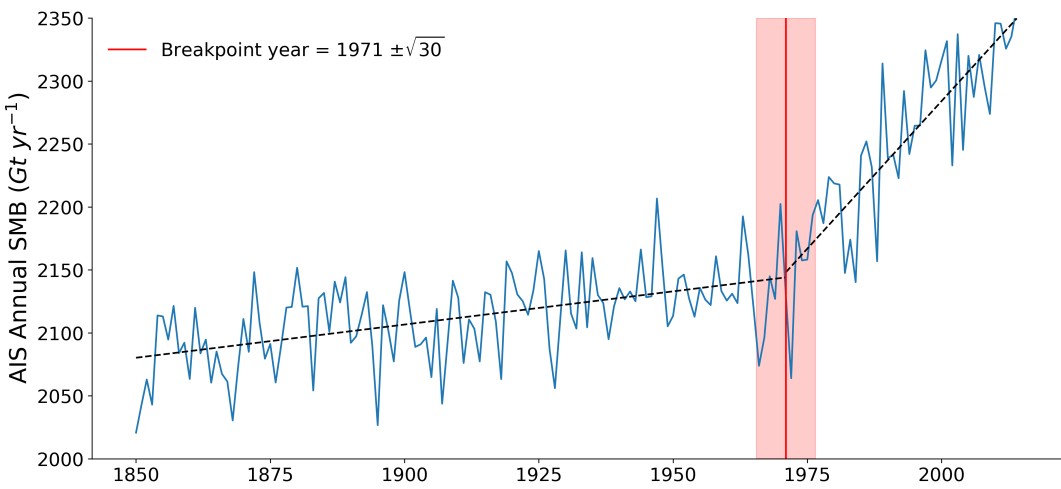

**Figure A6.** SMB breakpoint year, indicating the year with the greatest SMB change between preceding and subsequent 30 year periods. Uncertainty, shaded in red is defined as $\sqrt{n}$, which in this case is $\sqrt{30}$.

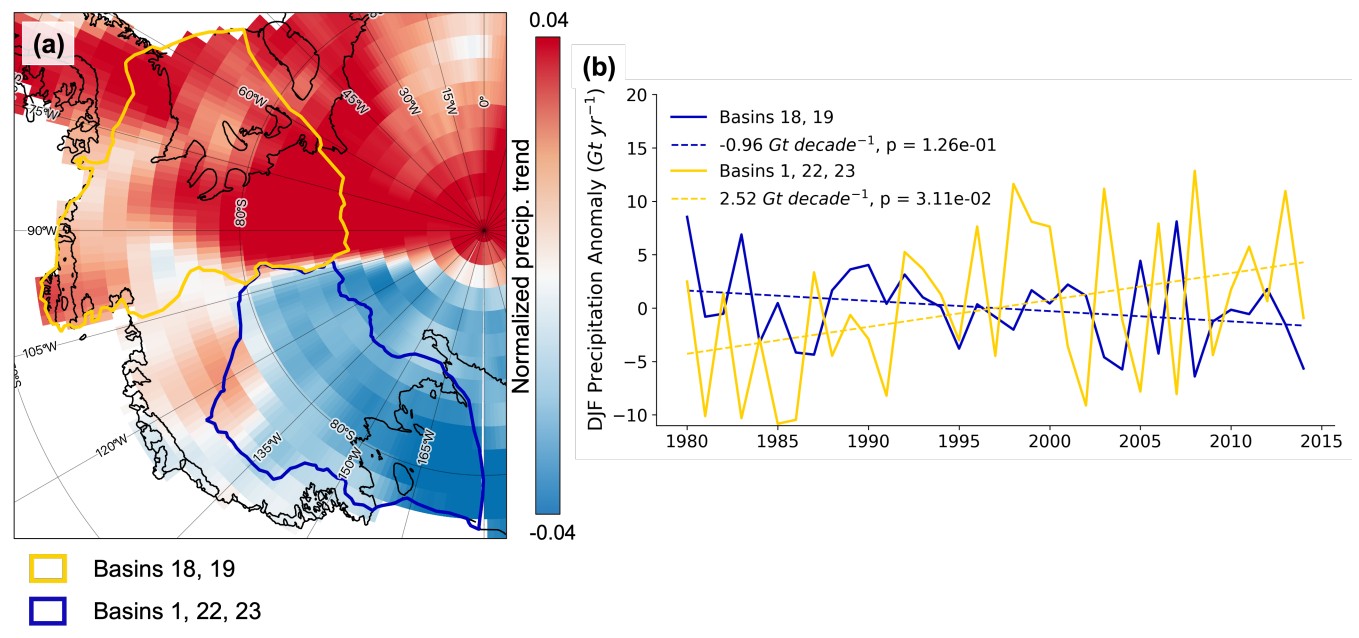

**Figure A7.** (a) Normalized 1979-2015 DJF trend in total precipitation from CESM2 with basins 18 and 19 outlined in blue and basins 1, 22, and 23 outlined in yellow. (b) Timeseries of yearly DJF precipitation anomaly with trend lines for areas outlined in (a).

## A1 Future model results

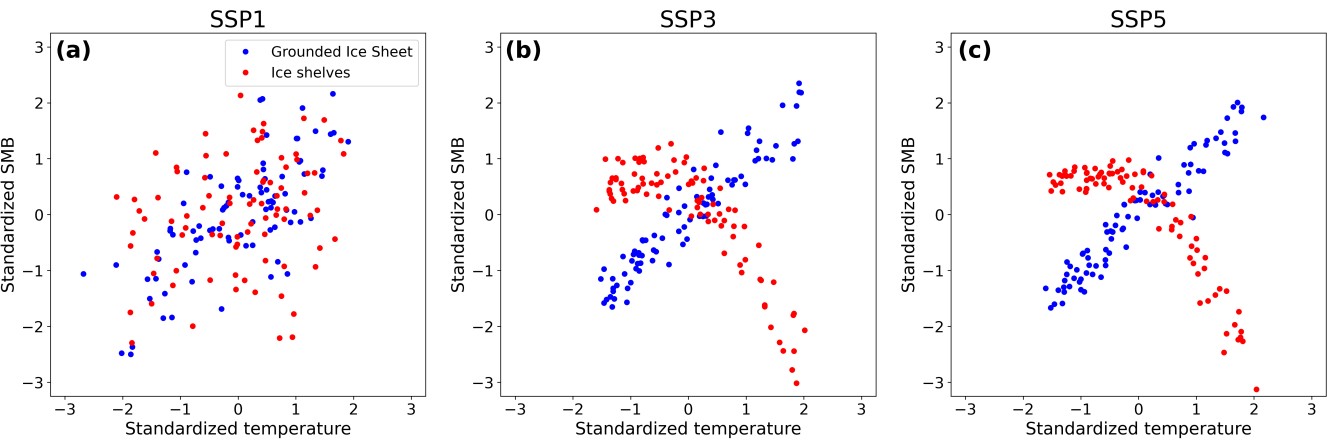

**Figure A8.** Annual average standardized temperature vs. annual average standardize SMB over ice shelves and the grounded ice sheet for every year from 2015-2100 from (a) SSP1-2.6, (b) SSP3-7.0, and (c) SSP5-8.5

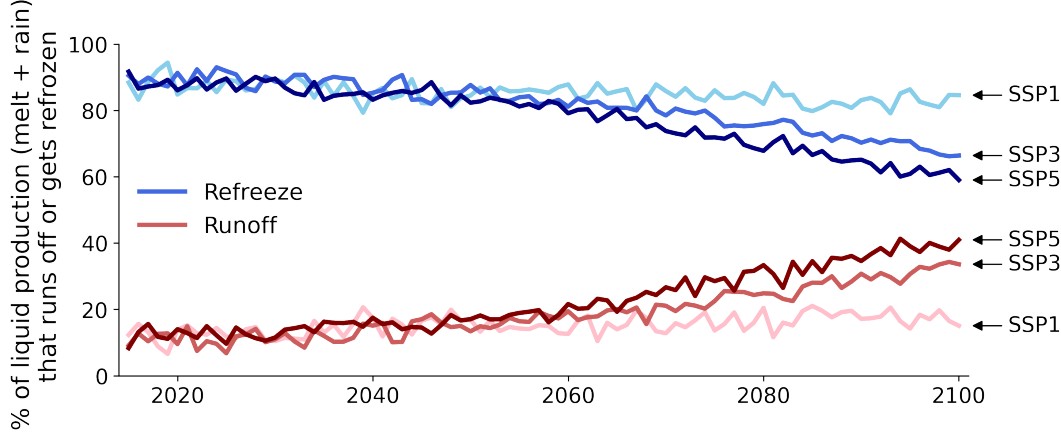

**Figure A9.** The percent of total AIS liquid production (melt + rainfall) that runs off (red) or gets refrozen (blue) in each future emission scenario.

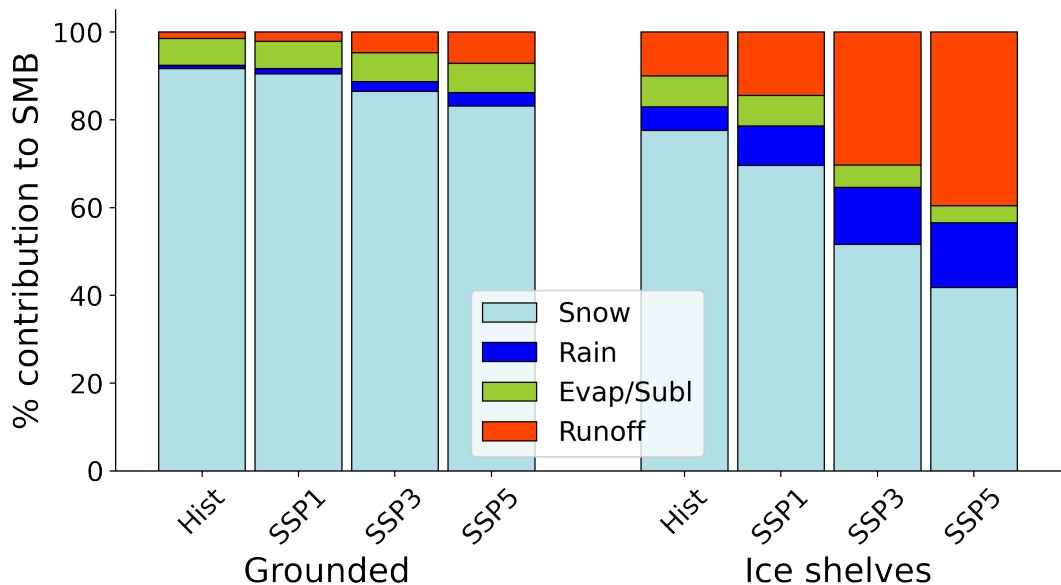

**Figure A10.** The contribution of snowfall, rainfall, evaporation/sublimation, and runoff to the total CESM2 SMB signal over the grounded ice sheet (left) and ice shelves (right) at the end of the historical period (2005-2015) and at the end of future scenarios SSP1-2.6, SSP3-7.0, and SSP5-8.5 (2090-2100).

*Data availability.* The QuikSCAT surface melt (Trusel et al., 2013) and RACMO2.3 SMB (van Wessem et al., 2017) products used in this study are a part of Quantarctica which can be downloaded at https://www.npolar.no/quantarctica/#toggle-id-15. The MAR SMB product is: year-MAR_ERA5-1979-2019_zen.nc2 and can be found at https://doi.org/10.5281/zenodo.4459259 (Kittel et al., 2021). The MERRA2

reconstruction (Medley and Thomas, 2019) product can be found at
https://earth.gsfc.nasa.gov/index.php/cryo/data/antarctic-accumulation-reconstructions. AWS observation data (Gossart et al., 2019) can be found at https://doi.org/10.5281/zenodo.6309896. ERA5 reanalysis output can be downloaded at https://cds.climate.copernicus.eu/#!/search? text=ERA5&type=dataset. Information about data from the CESM Large Ensemble project (Kay et al., 2015; Rodgers et al., 2021) can be found at https://www.cesm.ucar.edu/projects/community-projects/LENS/data-sets.html and CESM2 CMIP6 data can be found at https:

//esgf-node.llnl.gov/projects/cmip6/

.

*Code availability.* Code used to analyze all model output and make all figures in this manuscript can be found at https://github.com/ drdunmire1417/CESM2_analysis.

*Author contributions.* JL conceived of the study. Data collection was done by DD, JL, and TG, and analysis was done primarily by DD with help from JL and RTD. All authors contributed to the writing and editing of the manuscript.

*Competing interests.* The authors declare that they have no conflict of interest.

*Acknowledgements.* We acknowledge the CESM1 and CESM2 Large Ensemble Community Projects and supercomputing resources provided by NSF/CISL/Yellowstone, NSF/CISL/Cheyenne, and the IBS Center for Climate Physics in South Korea. We also thank Niels Souverijns and Alexandra Gossart for consolidating the AWS dataset. Finally, we would like the Editor, Nicolas Jourdain, and reviewers Christoph Kittel and another anonymous reviewer for their helpful comments which improved the quality of this manuscript.

*Financial support.* DD acknowledges support from NASA FINESST (grant 80NSSC19K1329). DD, JTML, and RTD were supported by the National Science Foundation, award 1952199.

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
