# Peer review of "Antarctic surface climate and surface mass balance in the Community Earth System Model version 2 during the satellite era and into the future (1979-2100)"

_The Cryosphere, 2022_

## Referee Comment (RC1)

*Review of 'Antarctic surface climate and surface mass balance in the Community Earth System Model 2 (1850-2100) by Dunmire et al. 2022'*

Dunmire et al. exhaustively evaluate the results of CESM2 over the Antarctic Ice Sheet against AWS observations and several often-used products such as RACMO, MAR, ERA5 or the reconstruction from Medley and Thomas (2019). The comparison is honest by highlighting both remaining biases and improvements against the previous version of the model. Furthermore, they discuss the future evolution of SMB using three different scenarios as it is a key variable for the ice sheet dynamics. One could argue than the topics is not particularly new, but this is an important study that would deserve a publication in the Cryosphere. CESM2 (one of the few polar-oriented ESMs) is often used either directly or downscalled to study the Antarctic climate and force ice sheet models. Knowing its biases is then important and I think that updates comparing to CESM1 (new emission scenarios, improved physics) are sufficient to justify a new study to present SMB projections. Furthermore, I found the paper well-structured, with clear figures and conclusions that are supported by results.

I have only minor comments listed hereafter that I hope will help the authors to improve their manuscript before its potential publication.

**Minor comments**

**P4 Section 2.5:** I recommend to specify the forcing of the RCM s(I think ERA-Interim for RACMO, ERA5 for MAR). Furthermore, you used ERA5 as a reanalysis but the reconstruction based on MERRA2. Yet this choice is explained and fully justified by the better performance on the reconstruction with this specific reanalysis but I wonder if the different forcings (MERRA2, ERA-Interim, ERA5) could results in different trends and change your comparison with CESM2. Since you also used a collection of products that could all give you the same information, is there any reason why you selected ERA5? For instance, precipitation or temperature could be also compared with RACMO or MAR. Try to justify why you selected ERA5 for these comparisons.

**P11 L205**: According to Figure 6a, I guess the values are for the grounded ice sheet. What is the SMB of CESM2 over the ice shelves? Are you well using a common mask? if yes specify it to help the readers interpretation, if not I strongly recommend to do it to compare something similar. (Also true for CMIP5 and CMIP6 values in Fig6.). There seems to have no evaluation/comparison over the ice shelves while other products than "the reconstruction" could be used there. Since ice shelves are particularly important for the Antarctic mass balance, this should be corrected.

**Trends**: Why do you use normalized trends? I understand that it better highlights the importance of small changes over areas with low values (eg., SMB over the high plateau) but at the same time it masks the real changes. Importance of small changes that are significant can still be highlighted by dots or crosses as you did. For Figures 3b and 7c indicate in the caption what crosses represent.

Most CESM2 trends are compared against other products (which is really interesting), but not the melt trend. I'd suggest to perform a similar comparison or at least cite a study (eg. Kuipers Munneke et al., 2012) that presents melt trends

**P14 L246 and after:** SSP scenarios are only mentioned using their first category (SSP5 instead of SSP5-8.5). Since there are several under scenarios in each category, keep mentioning the full name to remain clear.

**Specific comments and stylistic suggestions**

**P1 L4** : maybe « climate models » in general is enough than only ESM.

**P1 L14** : I suggest to replace « a coupled Antarctic Ice Sheet » by an « coupled ice sheet model » as it's not the real AIS that will be integreaed into CESM3. This is only a suggestion which the authors can obviously accept or refuse.

**P1 L24** : I agree about the stronger regional warming over these regions but the references are not adequate. The mass losses in West Antarctica are mainly due to ocean warming and not to the atmosphere that the references refer to. Increasing air-temperatures are more likely to contribute to hydrofracturing over the AP and subsequent glacier speed-up, but this is still a small contribution against the total mass loss over these two regions. Please reformulate/change your references.

**P2 L30** : Consider to remove « Studies have shown that »

**P2 L32** : Barthel et al. 2020 do not discuss the SMB uncertainty.

**P2 L54** : Add a reference (Gorthe et al., 2020?)

**P3 and P4 (Section 2.1 and 2.2)**: Do you use a specific member for the comparison or also the average of the 11 members ?

**P4 L106-108** : I suggest to specify that the SMB of the RCMs (and CESM2?) also includes the runof.

**P4 L111-11**2 : « The » reconstruction is perhaps a little over-emphasized given that the other products  (MAR and RACMO) also give reconstructions. (Again a suggestion, feel free to take into account or not). I'd suggest to refer to something like "the SMB reconstruction of Medley and Thomas (2019)" (or any abbreviation like MT2019 reconstruction).

**P6 L132 and 136** : Consider to replace « affect » by « effect ».

**P7 L139-149** : Are the temperature trends in ERA5 reliable ? If I'm not mistaken, most evaluations (eg., Gossart et al., 2019) only assessed the mean climate and not the trends. I would like more discussion on the potential reasons for these differences. Perhaps just mentioning that CESM2 is not constrained would be enough. Do you have a simulation where CESM2 is constrained that you could also compare to ERA5 (or AWS if ERA5 is not reliable) ? (see also the minor comment about trends above)

**P11 L205-204**: It's confusing that CESM2 SMB is significantly greater than RACMO SMB (1997 Gt/yr) but not significantly greater than "the reconstruction" (1953 Gt/yr). I guess this come from the large variability in the reconstruction. Do you know why this variability is so large? Is the variability computed on the same period ( as all the other products have almost the same variability)?

**P13 L236-240**: Consider to divide the sentence in several  ones to make it clearer.

**P14 L246**: Specify if you're presenting temperatures over the (grounded or full) ice sheet or over the regions.

**P14 L247**: "the first ten years of the future scenario (2015-2025) to the final ten years of the scenario (1990-2100)" Is there a mistake for the second period? (Shouldn't be 2090-2100?). Note that changes are more often compared to a selected period over the historical period than over a "future" period (I mean by "future", after 2014 where the scenario is no more the "historical" concentrations). The choice of the period should be consistent with P14 L268. Furthermore, are 10 years representative of the climates of both the "historical/start of the future period" and the end of the century?

**P14 L250**: Could you explain these differences? Are they due to the inertia of the system?

**P14 L270**: This is a really interesting analysis. The negative SMB in summer for all the scenario suggests high runoff values and in general strong melt and melt ponds. Since runoff indicates remaining liquid water at the surface (sometimes considered to be a proxy of potential hydro-fracturing – Donat-Magnin et al., 2021; Gilbert and Kittel, 2021), this might suggest that even for the low-emission scenario, surface melt could lead to severe damages over the ice shelves and strongly contribute to their disintegration with large consequences for the ice sheet stability. Maybe you could discuss/mention this in your manuscript.

**Appendix** : Change the order of the figures to match their order of appearance in the manuscript.

**Figures (**clear and adapted. I particularly appreciated Fig6.) For Figures 3b and 7c indicate in the caption what crosses represent.

Reference in this review:

Donat-Magnin, M., Jourdain, N. C., Kittel, C., Agosta, C., Amory, C., Gallée, H., ... & Chekki, M. (2021). Future surface mass balance and surface melt in the Amundsen sector of the West Antarctic Ice Sheet. The Cryosphere, 15(2), 571-593.

Gilbert, E., & Kittel, C. (2021). Surface melt and runoff on Antarctic ice shelves at 1.5 C, 2 C, and 4 C of future warming. Geophysical Research Letters, 48(8), e2020GL091733.

Kuipers Munneke, P., Picard, G., Van Den Broeke, M. R., Lenaerts, J. T. M., & Van Meijgaard, E. (2012). Insignificant change in Antarctic snowmelt volume since 1979. Geophysical Research Letters, 39(1).

---

## Author Comment (AC1)

**Reviewer #1**

Dunmire et al. exhaustively evaluate the results of CESM2 over the Antarctic Ice Sheet against AWS observations and several often-used products such as RACMO, MAR, ERA5 or the reconstruction from Medley and Thomas (2019). The comparison is honest by highlighting both remaining biases and improvements against the previous version of the model. Furthermore, they discuss the future evolution of SMB using three different scenarios as it is a key variable for the ice sheet dynamics. One could argue than the topics is not particularly new, but this is an important study that would deserve a publication in the Cryosphere. CESM2 (one of the few polar-oriented ESMs) is often used either directly or downscaled to study the Antarctic climate and force ice sheet models. Knowing its biases is then important and I think that updates comparing to CESM1 (new emission scenarios, improved physics) are sufficient to justify a new study to present SMB projections. Furthermore, I found the paper well-structured, with clear figures and conclusions that are supported by results.

> We thank the reviewer for their positive and encouraging comments regarding our manuscript.

I have only minor comments listed hereafter that I hope will help the authors to improve their manuscript before its potential publication.

**Minor comments**

P4 Section 2.5: I recommend to specify the forcing of the RCMs (I think ERA-Interim for RACMO, ERA5 for MAR)

> Thanks for suggesting this. We will update lines 107-108 in Section 2.5 to read: "we compared CESM2 results to RCM output from the latest versions of RACMO2.3, which is forced with ERA-Interim (van Wessem et al., 2017), and MAR (version 3.11), which is forced with ERA5 (Kittel et al., 2021).

Furthermore, you used ERA5 as a reanalysis but the reconstruction based on MERRA2. Yet this choice is explained and fully justified by the better performance on the reconstruction with this specific reanalysis but I wonder if the different forcings (MERRA2, ERA-Interim, ERA5) could result in different trends and change your comparison with CESM2.

> With regard to the mean-state SMB for the grounded ice sheet, all three reconstructions from Medley and Thomas (2019) have insignificantly different (p > 0.05) average annual SMB and have similar spatial patterns of relative bias when compared to CESM2 (see below):

MERRA2 reconstruction annual SMB:          1788 Gt/yr
ERA-Interim reconstruction annual SMB:     1816 Gt/yr
CFSR reconstruction annual SMB:            1750 Gt/yr

[Figure]

With regard to SMB trends, all three reconstructions have an insignificant SMB trend from 1950 to 2000. We only use the reconstruction product to compare with the CESM2 mean-state SMB and trend in SMB over the historical period (not other variables such as temperature or precipitation). Thus, our choice in using the MERRA2 reconstruction product will not change the comparison with CESM2 and is fully justified in line 110: "In this study we used the MERRA-2 based SMB reconstruction as it most closely resembles observations (Medley and Thomas, 2019)."

Since you also used a collection of products that could all give you the same information, is there any reason why you selected ERA5? For instance, precipitation or temperature could be also compared with RACMO or MAR. Try to justify why you selected ERA5 for these comparisons.

The near-surface precipitation and temperature trends from ERA5, RACMO2.3, and MAR are very similar (see figures below). We will add the figures below to the appendix and justify our decision to use ERA for the comparison with CESM2 by adding the following to our methods section 2.2.3:

"We also compared the CESM2 trend in near-surface temperature and precipitation from 1979-2015 with that from ERA5. We used ERA5 for this comparison because (a) it is the latest reanalysis product, with updated model physics and the highest horizontal resolution, and (b) has similar near-surface temperature and precipitation trends to the RCMS used in this study (Fig. A1, A2). The ERA5 near-surface temperature trend is also consistent with observations (Zhu et al., 2021)"

[Figure]

**Fig. A1.** 1979-2015 seasonal trend in near-surface temperature from ERA5 (a), RACMO2.3 (b), and MAR(c).

[Figure]

**Fig. A2.** 1979-2015 seasonal trend in precipitation from ERA5 (a), RACMO2.3 (b), and MAR(c).

P11 L205: According to Figure 6a, I guess the values are for the grounded ice sheet. What is the SMB of CESM2 over the ice shelves? There seems to have no evaluation/comparison over the ice shelves while other products than "the reconstruction" could be used there. Since ice shelves are particularly important for the Antarctic mass balance, this should be corrected.

> Figure 6 and lines 204-208 only include SMB over the grounded ice sheet because the reconstruction is only for the grounded ice sheet. We will clarify this in the text by changing lines ?? to "In CESM2, the annual average **grounded** surface mass balance (SMB) between 1979 and 2015 is...". We will also specify in the caption for Figure 6 that the timeseries (panel a) only includes the grounded ice sheet. To satisfy complaints from both reviewers we will expand our analysis in section 3.4.1 to include a comparison of ice shelf SMB from CESM2 with CESM1, ERA5, RACMO2.3, and MAR by adding the following paragraph:
>
> "Over ice shelves, CESM2 has an average SMB of 559 +/- 27 Gt $yr^{-1}$ between 1979 and 2015, significantly greater (p~<~0.05) than the average SMB over ice shelves from CESM1 (520 +/- 26 Gt $yr^{-1}$), ERA5 (506 +/- 26 Gt $yr^{-1}$), RACMO2.3 (523 +/- 24 Gt $yr^{-1}$), and MAR (459 +/- 23 Gt $yr^{-1}$). The MT2019 reconstruction only covers the grounded ice sheet and thus ice shelf SMB cannot be calculated from this product."

Are you well using a common mask? if yes, specify it to help the readers interpretation, if not I strongly recommend to do it to compare something similar. (Also true for CMIP5 and CMIP6 values in Fig6.).

We are using the Zwally mask which has been regridded for all of the modeling products used in this study. We will specify this and note the AIS grounded and ice shelf areas for each products in a new methods section 2.6 AIS Model masks:

"For area-integrated quantities we use the Zwally et al. (2012) AIS mask which has been re-gridded for all of the modeling products used in this study. The resulting grounded AIS areas from these models are as follows: 12043565 $km^2$ for CESM1 and CESM2, 12059084 $km^2$ for ERA5, 12063497 $km^2$ for RACMO2.3, 12154338 $km^2$ for MAR, and 12028208 $km^2$ for the MT2019 reconstruction. The resulting ice shelf areas from these models are: 1738581 $km^2$ for CESM1 and CESM2, 1755916 $km^2$ for ERA5, 1734991 $km^2$ for RACMO2.3, and 1749205 $km^2$ for MAR. Ice shelves are not included in the MT2019 reconstruction."

The annual AIS SMB values from all CMIP5 and CMIP6 models are from Gorte et al. 2020. The mask used for this analysis was the Ice Sheet Mass Balance Inter-comparison Exercise Team's (IMBIE Team) ice sheet mask.

Trends: Why do you use normalized trends? I understand that it better highlights the importance of small changes over areas with low values (eg., SMB over the high plateau) but at the same time it masks the real changes. Importance of small changes that are significant can still be highlighted by dots or crosses as you did. For Figures 3b and 7c indicate in the caption what crosses represent.

We agree that showing the normalized trends in Figure 7 is not necessary and will remove panel (c) from this figure. We will add cross-hatched areas to panel (b) to show areas where this trend is significant and add the following to Figures 3b and 7b: "Cross-hatched areas represent regions where this trend is significant (p < 0.05)."

Most CESM2 trends are compared against other products (which is really interesting), but not the melt trend. I'd suggest to perform a similar comparison or at least cite a study (eg. Kuipers Munneke et al., 2012) that presents melt trends.

We will cite the Kuipers Munneke et al., 2012 study to compare CESM2 historical melt trends with observations in the following sentence on line 193: "Historical (1979-2015) surface melt in CESM2 has increased across much of the AIS, a trend that is absent from observations (Kuipers Munneke et al., 2012)."

P14 L246 and after: SSP scenarios are only mentioned using their first category (SSP5 instead of SSP5-8.5). Since there are several under scenarios in each category, keep mentioning the full name to remain clear.

> We will update all references of SSP5 to SSP5-8.5, SSP3 to SSP3-7.0, and SSP1 to SSP1-2.6.

**Specific comments and stylistic suggestions**

P1 L4 : maybe « climate models » in general is enough than only ESM.

> We will keep ESM here because "climate models" could also include regional climate models or reanalysis climate models which we are not referring to in this line.

P1 L14 : I suggest to replace « a coupled Antarctic Ice Sheet » by an « coupled ice sheet model » as it's not the real AIS that will be integrated into CESM3. This is only a suggestion which the authors can obviously accept or refuse.

> We will change "coupled Antarctic Ice Sheet" to "coupled ice sheet model" on line 14.

P1 L24 : I agree about the stronger regional warming over these regions but the references are not adequate. The mass losses in West Antarctica are mainly due to ocean warming and not to the atmosphere that the references refer to. Increasing air-temperatures are more likely to contribute to hydrofracturing over the AP and subsequent glacier speed-up, but this is still a small contribution against the total mass loss over these two regions. Please reformulate/change your references.

> We will change the sentence in L24 to read: "Ice shelves in the Amundsen and Bellingshausen sea regions are thinning in large part due to increased basal melting (Pritchard et al., 2012), a process that reduces the buttressing effect of ice shelves and leads to increased ice discharge (Rignot et al., 2019, Milillo et al., 2022)"

P2 L30 : Consider to remove « Studies have shown that »

> We will remove the words "Studies have shown that" in line 30 such that this sentence now begins: "Historical increases in AIS SMB…"

P2 L32 : Barthel et al. 2020 do not discuss the SMB uncertainty.

> We will remove the "Barthel et al. 2020" citation in line 32.

P2 L54 : Add a reference (Gorthe et al., 2020?)

> We will add the "Gorte et al. (2020)" citation to the sentence "CMIP6 modeled annual SMB values… mean of 2127 Gt yr$^{-1}$." in line 54.

P3 and P4 (Section 2.1 and 2.2): Do you use a specific member for the comparison or also the average of the 11 members?

We will clarify that we use the average of the 11 ensemble members by stating in section 2.1.1: "For comparing the CESM2 mean and uncertainty of these output variables to other products we calculated the 11-member ensemble average mean and standard deviation."

P4 L106-108 : I suggest to specify that the SMB of the RCMs (and CESM2?) also includes the runoff.

Runoff is included in SMB calculations from all model simulations except for ERA5. We will update the ERA5 SMB calculation to include runoff and will clarify this by changing line 106 to "(approximated by precipitation - evaporation/sublimation - runoff)"

P4 L111-112 : « The » reconstruction is perhaps a little over-emphasized given that the other products (MAR and RACMO) also give reconstructions. (Again a suggestion, feel free to take into account or not). I'd suggest to refer to something like "the SMB reconstruction of Medley and Thomas (2019)" (or any abbreviation like MT2019 reconstruction).

We will change instances of "the reconstruction" to "the MT2019 reconstruction", which we will define in Methods section 2.5.

P6 L132 and 136 : Consider to replace « affect » by « effect ».

We will make  these changes to lines 132 and 136.

P7 L139-149 : Are the temperature trends in ERA5 reliable? If I'm not mistaken, most evaluations (eg., Gossart et al., 2019) only assessed the mean climate and not the trends. I would like more discussion on the potential reasons for these differences. Perhaps just mentioning that CESM2 is not constrained would be enough. Do you have a simulation where CESM2 is constrained that you could also compare to ERA5 (or AWS if ERA5 is not reliable)? (see also the minor comment about trends above)

ERA5 temperature trends are generally deemed reliable in those regions where we have long-term, high quality temperature observations (Zhu et al., 2021). As the reviewer indicates, CESM2 is fully coupled so is not constrained by any temperature observations. This will be clearly indicated in the Methods section of the text.

P11 L205-204: It's confusing that CESM2 SMB is significantly greater than RACMO SMB (1997 Gt/yr) but not significantly greater than "the reconstruction" (1953 Gt/yr). I guess this come from the large variability in the reconstruction. Do you know why this variability is so large? Is the variability computed on the same period (as all the other products have almost the same variability)?

We have checked our significance tests for the SMB difference between CESM2 and the various other products and found that CESM2 SMB is significantly greater than all products (including MAR, for which we recalculated SMB using the Zwally mask used for all other products, and the reconstruction). For the reconstruction, the "variability" is so large because here we use the reconstruction error as provided by Medley and Thomas, 2019. We will clarify this at the end of section 2.5 by adding the following sentence:

"We use the SMB error provided by Medley and Thomas (2019) as the variability for this dataset."

P13 L236-240: Consider to divide the sentence in several ones to make it clearer.

We will split this sentence into 2 separate sentences so that it reads: "Differences in historical precipitation trend between ERA5 and CESM2 exist across much of the AIS, but particularly in Wilkes Land and Princess Elizabeth Land ($\sim$75 °E – 136 °E), with precipitation largely decreasing in ERA5 but increasing in CESM. Additionally, over the eastern AP ($\sim$63 °W) in DJF, precipitation decreases strongly in ERA5 but remains roughly constant in CESM2."

P14 L246: Specify if you're presenting temperatures over the (grounded or full) ice sheet or over the regions.

We will specify that we are presenting temperature over the full ice sheet in line 246.

P14 L247: "the first ten years of the future scenario (2015-2025) to the final ten years of the scenario (1990-2100)" Is there a mistake for the second period? (Shouldn't be 2090-2100?).

Correct, it should be 2090-2100. We will fix this mistake in line 247 of the manuscript.

Note that changes are more often compared to a selected period over the historical period than over a "future" period (I mean by "future", after 2014 where the scenario is no more the "historical" concentrations). The choice of the period should be consistent with P14 L268.

For the calculation of the $\frac{\partial SMB}{\partial T}$ values presented in section 3.5 we will change the period to be consistent with P14 L268 such that we are comparing the final 10 years of the historical simulation (2005-2015) with the final 10 years of the future simulation (2090-2100).

Furthermore, are 10 years representative of the climates of both the "historical/start of the future period" and the end of the century?

The objective of this analysis and of Figure 9 is to capture the change in SMB over the future period. We are not comparing representative climates because the climate is changing so rapidly throughout the future period, even within 10 years. We chose to use 10 years because it is a long enough period to account for inter-annual variability and allows us to evaluate the change in SMB over the future period in different scenarios.

P14 L250: Could you explain these differences? Are they due to the inertia of the system?

We believe that the diverging SMB trend on ice shelves contributes to the differences in $\frac{\partial SMB}{\partial T}$ between the different SSP scenarios. These rate change numbers are for the full ice sheet, something we have clarified on line 246. A strongly decreasing ice shelf SMB in SSP5-8.5 contributes to a lower rate of SMB change with increasing temperatures. This is stated on line 253-254: "A diverging future SMB trend on ice shelves and the grounded ice sheet, of which CESM2 agrees with previous studies (Kittel et al., 2021), is responsible for the varying $\frac{\partial SMB}{\partial T}$ between different emission scenarios."

P14 L270: This is a really interesting analysis. The negative SMB in summer for all the scenario suggests high runoff values and in general strong melt and melt ponds. Since runoff indicates remaining liquid water at the surface (sometimes considered to be a proxy of potential hydrofracturing– Donat-Magnin et al., 2021; Gilbert and Kittel, 2021), this might suggest that even for the low-emission scenario, surface melt could lead to severe damages over the ice shelves and strongly contribute to their disintegration with large consequences for the ice sheet stability. Maybe you could discuss/mention this in your manuscript.

Given the limitations in CESM2's ability to simulate meltwater processes, we would like to refrain from making substantial claims on ice shelf vulnerability to hydrofracture, and stick with the analysis of SMB and components. Surface melt and processes related to surface melt (ie refreezing, liquid water storage at the surface) are still not well captured by CESM2. Additionally, there are lots of other factors that contribute to meltwater ponding and the potential for hydrofracture (i.e. the availability of near-surface firn, the formation of impermeable ice lenses) that are likely not captured by CESM2. We propose that atmospheric forcing from CESM2 would need to be used as input for a more sophisticated firn model to analyze the impacts on ice shelf surface hydrology.

Appendix: Change the order of the figures to match their order of appearance in the manuscript.

We will update the order of the appendix figures to match their reference in the manuscript.

Figures (clear and adapted. I particularly appreciated Fig6.) For Figures 3b and 7c indicate in the caption what crosses represent.

> Thank you, we will indicate in the captions of Figures 3 and 7 what cross-hatching represents.

**References**

Gorte, T., Lenaerts, J. T., and Medley, B.: Scoring Antarctic surface mass balance in climate models to refine future projections, Cryosphere, 14, 4719–4733, https://doi.org/10.5194/tc-14-4719-2020, 2020.

Kuipers Munneke, P., G. Picard, M. R. van den Broeke, J. T. M. Lenaerts, and E. van Meijgaard: Insignificant change in Antarctic snowmelt volume since 1979, Geophysical Research Letters., 39, L01501, doi:10.1029/2011GL050207, 2012.

Milillo, P., Rignot, E., Rizzoli, P., et al: Rapid glacier retreat rates observed in West Antarctica, Nature Geoscience, 15, 48-53, 2022.

Pritchard, H.D., Ligtenberg, S. R. M., Fricker, H. A., et al: Antarctic ice-sheet loss driven by basal melting of ice shelves, Nature, 484, 502-505, 2012.

Zhu, J., Xie, A., Qin, X., et al.: An assessment of ERA5 reanalysis for antarctic near-surface air temperature, Atmosphere, 12, 217, 2021.

---

## Author Comment (AC2)

**Reviewer #2**

**Summary**

This paper describes the performance of the CESM2 model in simulating factors relevant to the surface mass balance of the Antarctic ice sheet in the recent past and the next century. It is a useful exercise, providing documentation and detail for people who use CESM2 or its output to force ice sheet models, for other ESM groups as an example of good evaluation practice and context, and for cryosphere scientists who might not know what to expect of a global ESM simulation of AIS climate. As primarily a descriptive paper it doesn't contain revolutionary science conclusions, but it is still a very worthwhile contribution and I would consider it a TC paper rather than model development that might be more suited to e.g. Geoscientific Model Development. It is well written and organized with a good level of detail. I would recommend publishing it with minor revisions.

> We thank the reviewer for their positive and encouraging comments regarding our manuscript.

**General comments**

Some suggestions to consider for those minor revisions that would improve the utility of the paper:

Obviously, this paper focuses mainly on factors local to the AIS surface, but would also be nice to see a little more assessment of the regional climate away from AIS (confer the metrics used in assessing GCM climate plausibility for AIS SMB in section 2.2 of Barthel et al. 2020, https://doi.org/10.5194/tc-14-855-2020) that will have a significant influence on e.g. the amount of precip that gets tracked in from the ocean, or warm air advection. It would be good to have an attempt at attributing improvements wrt CESM1 to either the large-scale CESM2 climate or the changes in surface modelling.

> We agree with the reviewer that it is important to relate our findings to other climate processes in and around the AIS. There are several studies that provide an assessment of the regional climate away from the AIS and the large-scale CESM2 climate (Simpson et al 2020, Singh et al 2021, Raphael et al 2020, Dalaiden et al 2020). We argue that these studies provide sufficient analysis of CESM2's performance in reproducing the high-latitude climate system above and around Antarctica. To better link our results to this existing body of literature, we will add the following paragraph to the discussion section:
>
> "In the context of the larger Southern Hemisphere (SH), Delaiden et al (2020) show that the CESM2 Antarctic moisture budget due to synoptic and large-scale atmospheric circulation is realistic compared to reanalysis (ERA-Interim). This indicates that unrealistic CESM2 mean-state precipitation may be attributed to

cloud microphysics, not SH moisture budget. While CESM2 performs well regarding the mean-state SAM and the location of the SH jet, its representation of stationary waves and the speed of the SH jet have degraded from CESM1 (Simspon et al. 2020). Zonal circulation appears overall too strong in CESM2, which may enhance or reduce precipitation in various regions across the AIS. Analogous to the unrealistic precipitation trend in CESM2, there is also a decrease in CESM2 SH sea ice throughout the historical period that cannot be reconciled with observations (Duvivier et al. 2020, Raphael et al. 2020). The unrealistic SH sea ice and AIS precipitation trends may arise from similar factors (i.e. high CESM2 climate sensitivity); and/or, a decrease in sea ice may contribute to increasing AIS precipitation."

Whatever the cause, the improvements in match to the reference products at the end of the 20th century wrt CESM1 are very encouraging, and that - along with the potential for improving the sea-level projection capabilities in CMIP that are mentioned - should definitely be highlighted. The flipside though, whose impact I think could be discussed more and might carry as much importance, is what appears to be a very significant discrepancy in the sensitivity of the simulated precipitation to changes in 20th century climate. What credence should users of this model/simulation data put in the projections of future SMB on that evidence? I'm not criticizing the simulation, just asking for a more prominent discussion of the implications. I've just read the paper on the PaleoCalibr version of CESM2 (Zhu et al 2021, https://doi.org/10.1029/2021MS002776) which claims to have a better/more physical tuning of nucleation around ice in clouds and affects high latitude cloud and general climate sensitivity significantly. Since these cloud microphysics are mentioned as important in this text, it might be nice to have a comment on what, if any, impact they think that tuning might have on their SMB simulation and sensitivity?

> We will add the following (after line 296) to our discussion to address unrealistic CESM2 climate sensitivity as noted by the reviewer:
>
> "Zhu et al. (2021) find that the CESM2 climate is very sensitive to treatments of cloud microphysical processes and that tuning these processes results in a modeled climate sensitivity that more realistically matches present-day observations. CESM2's unrealistically high climate sensitivity likely implies that modeled future precipitation and runoff trends are also overestimated, something that should be taken into consideration when discussing AIS SMB under different future emissions scenarios in CESM2."

The title promises 1850-2100 but I think this is a little misleading:

1) The focus is mostly on evaluation against various high resolution or observationally-based products, so for understandable reasons there's virtually nothing about the situation before the 1970s. The only exception is figure B2, used to establish the change in precipitation trend in the model at the end of the 20th century, but that's a fairly scant use for 120+ years of

simulation.  There is presumably little that can be said other than presenting the differences with CESM1 for the earlier period - that might be interesting in itself, or it might be better to change the title.

> We agree with the reviewer that the focus is mostly on evaluating the model following 1979 when satellite-based products and observations were available. We will change the manuscript title to: "Antarctic surface climate and surface mass balance in the Community Earth System Model 2 during the satellite era and into the future (1979-2100)" to be more in-line with the topics of this paper.

2) Everything after 2014, for all climate and surface variables across several different scenarios, is condensed into 1 side of text and 2 figures - the future feels pretty shortchanged as well. What is there is good but it would be nice to see much more detail. For instance, does the pattern of increase in precipitation just match surface temperatures and scale with Clausius-Clapeyron humidity arguments, or is there evidence of something dynamic eg more frequent/vigorous storm incursions. How do the areas of increased shelf melt compare to estimates of which shelves will be vulnerable to hydrofracture from surface melting in the future. eg Kuipers Munneke et al. 2017 (https://doi.org/10.3189/2014JoG13J183).

> An analysis of increased precipitation based on thermodynamical vs dynamical impacts has been done by Delaiden et al (2020) (see Figure 3) and a more thorough investigation of dynamics-induced precipitation changes is beyond the scope of this paper. We agree that Section 3.5 on future model trends should be expanded and will add the following paragraph and corresponding supplemental figure:

> "At the end of the historical simulation (2005-2015), solid precipitation contributes to 91.7% of the total grounded SMB signal in CESM2, while rainfall, evaporation/sublimation, and runoff contribute 0.7%, 6.1%, and 1.5% respectively (Fig. A8). By the end of the future period (2090-2100), the contribution of both rainfall and runoff to the modeled SMB signal increases slightly in all scenarios (3.1% and 7.1%, respectively in SSP5-8.5), with a corresponding decrease in the contribution of precipitation (83.1% in SSP5-8.5). Over ice shelves, we see a much greater change in the contribution of these different components to the total CESM2 SMB signal at the end of the future period (Fig. A8). From 2005 to 2015, snowfall accounts for 77.6% of the modeled ice shelf SMB signal, rainfall accounts for 5.4%, evaporation/sublimation accounts for 7.0%, and runoff accounts for 10.0%. By the end of the SSP5-8.5 scenario, snowfall accounts for less than half of the ice shelf SMB signal (41.8%), with rainfall, evaporation/sublimation, and runoff accounting for 14.8%, 3.9%, and 39.5%, respectively."

[Figure]

**Fig. A8.** The contribution of snowfall, rainfall, evaporation/sublimation, and runoff to the total CESM2 SMB signal over the grounded ice sheet (left) and ice shelves (right) at the end of the historical period (2005-2015) and at the end of future scenarios SSP1-2.6, SSP3-7.0, and SSP5-8.5 (2090-2100).

Given the limitations in CESM2's ability to simulate meltwater processes, we would like to refrain from making substantial claims on future ice shelf vulnerability to hydrofracture, and stick with the analysis of SMB and components. Surface melt and processes related to surface melt (ie refreezing, liquid water storage at the surface) are still not well captured by CESM2. Additionally, there are lots of other factors that contribute to meltwater ponding and the potential for hydrofracture (i.e. the availability of near-surface firn, the formation of impermeable ice lenses) that are likely not captured by CESM2. We propose that atmospheric forcing from CESM2 would need to be used as input for a more sophisticated firn model to analyze the impacts on ice shelf surface hydrology.

**Specific Comments:**

Line 1, abstract: could be more concise and to the point. I'm not sure the first or last 3 sentences are really needed here at all, since they provide general information and broad justification that could live in the Introduction. Some more numbers - integrated SMB, precipitation, runoff, percentage discrepancy from the reference products in the present, projections for the future - would be more useful. I would note the issue with the historical trends in CESM2 here too, as well as the marked improvement in the mean state wrt CESM1.

We will trim the first and last 3 sentences of the abstract to be more concise. We will also add a sentence regarding the issue with the historical trends in CESM2. Our modified abstract will read:

"Earth System Models (ESMs) allow us to explore minimally-observed components of the Antarctic Ice Sheet (AIS) climate system, both historically and under future climate change scenarios. Here, we present and analyze surface climate output from the most recent version of the National Center for Atmospheric Research's ESM: the Community Earth System Model version 2 (CESM2). We compare AIS surface climate and surface mass balance (SMB) trends as simulated by CESM2 with reanalysis and regional climate models and observations. We find that CESM2 substantially better represents the mean state of AIS near-surface temperature, wind speed, and surface melt compared with its predecessor, CESM1. This improvement is likely a result of the inclusion of new cloud microphysical parameterizations and changes made to the snow model component. However, we also find that grounded CESM2 SMB (2269 +\- 100 Gt/yr) is significantly higher than all other products used in this study and that both temperature and precipitation are increasing across the AIS during the historical period, a trend that cannot be reconciled with observations. This study provides a comprehensive analysis of the strengths and weaknesses of CESM2 representation of AIS surface climate, which will be especially useful in preparation for CESM3, which plans to incorporate a coupled ice sheet model that interacts with the ocean and atmosphere. "

Line 23: "attributing to" - should be "accounting for", perhaps?

We will change "attributing to" to "accounting for" in line 23 of the manuscript.

Line 40: the word "limit..." gets used a lot in these two sentences

We will update these sentences in lines 40-42 to read "Because of Antarctica's remoteness, in-situ observations are spatially and temporally sparse, limiting our understanding of how the surface climate and SMB are changing. Accordingly, we use additional products to assess the AIS surface climate, each with its own set of advantages and disadvantages."

Line 79, section2.1.2: are the CESM2 mec subgridscale ice elevation classes used for the AIS in either version? I think there were particular parameterisation tunings that were done for the calculation of Greenland SMB in CESM2, at least for the version of the model with interactive ice - eg adjustments to the phase of precipitation for certain land surface type (van Kampenhout et al, 2020 https://doi.org/10.1029/2019JF005318) - are they active here?

Yes, CESM has MECs active over Antarctica. Since the downscaling does not change the grid cell integrated mass or energy fluxes, CESM2 is not coupled to an

> ice sheet model over the AIS, and most atmospheric variables are not downscaled, we decided to present our results on the native CESM2 grid. We will make this clear in the Methods section in the manuscript.

Line 106: I may have missed something, but is this approximation for SMB used consistently in the analysis of all the model simulations and evaluation products, or does it only apply to ERA? If this is the formal definition to be used in the paper, it then needs amending for the future results where runoff becomes very significant.

> Runoff is included in SMB calculations from all model simulations except for ERA5. We will update the ERA5 SMB calculation to include runoff and will clarify this by changing line 106 to "(approximated by precipitation - evaporation/sublimation - runoff)"

Figure 1: Does panel(d) impart useful information that can't be got from panel(e)?

> We agree that panel d from Figures 1 and 4 are not necessary and will remove them from the Figures.

Line 137: can the CESM radiation components shown (and other surface energy fluxes) be compared with the RCMs/reanalysis?

> We will add a comparison of incoming short and longwave radiation from CESM2 with ERA5 to Figure 2 (below) and the following paragraph to the results section:
>
> "Compared with ERA5 (Fig. 2c,f), CESM2 has a spatially-averaged -7.3 W m$^{-2}$ bias in incoming shortwave radiation (an improvement from the +20.8 W m$^{-2}$ CESM1 bias) and a -1.8 W m$^{-2}$ bias in incoming longwave radiation (improved from -12.2 W m$^{-2}$ in CESM1). ERA5 suggests that CESM2 incoming shortwave radiation is negatively biased at the AIS coast and positively biased in the interior (Fig. 2c), a spatial pattern that is consistent with CESM2 near-surface temperature biases whereby modeled temperatures are largely too cold along the coast and too warm in the interior (Fig. 1c)."

[Figure]

**Figure 2.** Comparison of incoming radiation components between CESM2 (1979-2015), CESM1 (1979-2005) and ERA5 (1979-2015). (a) CESM2 average austral summer incoming shortwave radiation. (b) CESM2 - CESM1 average austral summer incoming shortwave radiation. (c) CESM - ERA5 average austral summer incoming shortwave radiation. (d) CESM2 average annual incoming longwave radiation. (e) CESM2 - CESM1 average annual incoming longwave radiation. (f) CESM2 - ERA5 average annual incoming longwave radiation.

We will also update appendix Figure B1 to include a comparison of the sensible and latent heat flux between CESM2 and ERA5:

[Figure]

**Figure B1.** Comparison of turbulent fluxes between CESM2 (1979-2015), CESM1 (1979-2005) and ERA5 (1979-2015). (a) CESM2 average annual latent heat flux (LHF). (b) CESM2 - CESM1 average annual LHF. (c) CESM - ERA5 average annual LHF. (d) CESM2 average annual sensible heat flux (SHF) (e) CESM2 - CESM1 average annual SHF. (f) CESM2 - ERA5 average annual SHF. Positive values indicate a downward net energy flux (into the ice sheet).

Line 152: are the katabatic winds really well-resolved in a 1degree model?

> Our results show that CESM2 well-represents these strong down-slope winds in East Antarctica, so yes, these katabatic, or downslope winds appear to be resolved in a 1 degree model.

Line 169: these are the first of the area-integrated quantities, I think. It would be worth saying what the size of the AIS is in each of these different models/products. It's not obvious that a 1degree GCM will represent the AIS with exactly the same extent as a more detailed regional model, a factor which might bias results systematically high or low.

> We are using the Zwally mask which has been regridded for all of the modeling products used in this study. We will specify this and note the AIS grounded and ice shelf areas for each products in a new methods section 2.6 AIS Model masks:
>
> "For area-integrated quantities we use the Zwally et al. (2012) AIS mask which has been re-gridded for all of the modeling products used in this study. The resulting grounded AIS areas from these models are as follows: 12043565 km$^2$ for CESM1 and CESM2, 12059084 km$^2$ for ERA5, 12063497 km$^2$ for RACMO2.3, 12154338 km$^2$ for MAR, and 12028208 km$^2$ for the MT2019 reconstruction. The resulting ice shelf areas from these models are: 1738581 km$^2$ for CESM1 and CESM2, 1755916 km$^2$ for ERA5, 1734991 km$^2$ for RACMO2.3, and 1749205 km$^2$ for MAR. Ice shelves are not included in the MT2019 reconstruction."

Line 182: "to too melt" - too /much/ melt?

> We will change the phrase "to too melt" to "too much melt" on line 182.

Line 202, section 3.4: I know accumulation vastly outweighs other things in the current SMB balance, but we're not actually told a split between accumulation and ablation (whether or not that includes runoff) at any point here? Could something be shown on sublimation? Section 3.3 talks about surface melt, but no note of whether any of that runs off - the runoff and refreeze proportion becomes important in the future section later, so I think it should be mentioned for the present-day too, even if only briefly.

> We will add the following paragraph in Section 3.4.1 to briefly quantify the contribution of accumulation and ablation terms in the total SMB signal:

"For the full ice sheet, accumulation from both solid and liquid precipitation accounts for 91.7% of the total SMB signal in CESM2, with ablation terms accounting for 8.3% of the signal (6.5% from sublimation/evaporation and 1.8% from runoff). This breakdown is comparable to that from ERA5, where 92.1% of the total SMB signal comes from precipitation, 6.9% from sublimation/evaporation, and 1.0% from runoff. In comparison, only 2.0% of the total SMB signal from CESM1 comes from sublimation/evaporation (with 96.6% from precipitation and 1.4% from runoff). This increase in the sublimation/evaporation contribution to the SMB signal from CESM1 to CESM2 is likely due to the increase in near-surface wind speed (discussed in Section 3.2) which drives a corresponding increase in latent heat flux between the model versions."

Line 231: it could be clearer whether these two sentences are talking about CESM2 or reality.

We will update the two sentences in line 231 to read: "AIS historical precipitation trends in CESM2 appear to be largely driven by the increasing SAM and intensifying Antarctic ozone depletion, with spatial patterns similar to that shown in Lenaerts et al. (2018)."

Figure 6(b,c,d): it's not clear what's going on with the ice shelves. Are they excluded from the SMB analysis, and we're only talking about grounded ice here? Is it an issue with just this figure, or throughout the paper? Figure 5 shows melt on the shelves, so they're not excluded from all the analysis

Thank you for pointing out this confusion. Figure 6 and lines 204-208 only include SMB over the grounded ice sheet because the reconstruction only covers the grounded ice sheet. We will clarify this in the text by changing line 204 to "In CESM2, the annual average **grounded** surface mass balance (SMB) between 1979 and 2015 is…". We will also specify in the caption for Figure 6 that the timeseries (panel a) only includes the grounded ice sheet. To satisfy complaints from both reviewers we will expand our analysis in section 3.4.1 to include a comparison of ice shelf SMB from CESM2 with CESM1, ERA5, RACMO2.3, and MAR by adding the following paragraph:

"Over ice shelves, CESM2 has an average SMB of 559 +/- 27 Gt yr$^{-1}$ between 1979 and 2015, significantly greater (p < 0.05) than the average SMB over ice shelves from CESM1 (520 +/- 26 Gt yr$^{-1}$), ERA5 (506 +/- 26 Gt yr$^{-1}$), RACMO2.3 (523 +/- 24 Gt yr$^{-1}$), and MAR (459 +/- 23 Gt yr$^{-1}$). The reconstruction only covers the grounded ice sheet and thus ice shelf SMB cannot be calculated from this product."

We will also included ice shelf SMB from CESM2 in Figure 6 panel b to indicate that our analysis does include ice shelves in this manuscript.

Line 242, section 3.5: is there a general "future simulations with CESM2" paper that could be cited for more context (eg Meehl et al., https://doi.org/10.1029/2020EA001296)?

Thanks for the suggestion, we will add the Meehl et al (2020) citation to the sentence in line 242.

Line 246: "SSPx" is an incomplete reference to which emissions scenario is being talked about - eg SSP5-8.5

We will update all references of SSP5 to SSP5-8.5, SSP3 to SSP3-7.0, and SSP1 to SSP1-2.6.

Line 247: "1990" is probably a typo

Thanks for catching this, we will update this typo to: (2090 – 2100) in line 247.

Line 257: is there any evidence from the 20th century simulation that the available pore space and refreezing in CESM2 are realistic to start with?

No, unfortunately there is no evidence that the available pore space and refreezing in CESM2 are realistic during the historical simulation. We discuss this limitation in lines 310-314 of the manuscript: "While CESM2's firn model has improved substantially (van Kampenhout et al., 2017), it still only allows for a ~20-30 meters deep firn column, which likely results in an underestimation of meltwater storage capacity in the firn across much of the AIS. In a future warming climate with non-linearly increasing meltwater production on Antarctic ice shelves, CESM2 may exaggerate runoff as a result of this shallow firn column, highlighting the need for a continued development of the snow model to better understand future SMB changes.

Line 315: "in the latest iteration of CMIP", perhaps - various EMICS, simpler models and even a CMIP6 ESM (Siahaan et al. 2021, https://doi.org/10.5194/tc-2021-371) have produced "coupled" estimates of future AIS contributions to sea level rise, for whatever they're worth.

We will update line 315 to read: "Even in the latest iteration of estimating future AIS contribution to sea level rise, few attempts have been made to couple ice sheet and ESMs (Siahaan et al., 2021). Antarctic ice sheet models are largely simulated as a stand-alone, meaning they require climate forcing (Seroussi et al., 2020)."

Line 316: "will be used" - it already has been (Payne et al. https://doi.org/10.1029/2020GL091741)!

We will change "will be used" in line 316 to "**will be more extensively used** as this forcing for ice sheet models (Payne et al. 2021)".

Line 324: I don't think the repeat of the factors in parentheses is required.

We will delete the text in parentheses in line 324.

**References**

Dalaiden, Q., Goosse, H., Lenaerts, J., et al. Future Antarctic snow accumulation trend is dominated by atmospheric synoptic-scale events, Communications Earth & Environment, 1, 1-9, 2020.

Raphael, M., Handcock, M., Holland, M., et al. An Assessment of the Temporal Variability in the Annual Cycle of Daily Antarctic Sea Ice in the NCAR Community Earth System Model, Version 2: A Comparison of the Historical Runs With Observations, Journal of Geophysical Research: Oceans, 125, 2020.

Payne, A., Nowicki, S., Abe-Ouchi, A., et al. Future Sea Level Change Under Coupled Model Intercomparison Project Phase 5 and Phase 6 Scenarios From the Greenland and Antarctic Ice Sheets, Geophysical Research Letters, 48, 1-8, 2021.

Siahaan, A., Smith, R., Holland, P., et al. The Antarctic contribution to 21st century sea-level rise predicted by the UK Earth System Model with an interactive ice sheet, The Cryosphere Discussions, 1-42, 2021.

Simpson, I., Bacmeister, J., Neale, R., et al. An Evaluation of the Large-Scale Atmospheric Circulation and Its Variability in CESM2 and Other CMIP Models, Journal of Geophysical Research: Atmospheres, 125, 1-42, 2020.

Singh, H., Landrum, L., Holland, M., et al. An Overview of Antarctic Sea Ice in the Community Earth System Model Version 2, Part 1: Analysis of the Seasonal Cycle in the Context of Sea Ice Thermodynamics and Coupled Atmosphere-Ocean-Ice Processes, Journal of Advances in Modeling Earth Systems, 3, 2021.

Zhu, J., Otto-Bliesner, B., Brady, E., et al. LGM Paleoclimate Constraints Inform Cloud Parameterizations and Equilibrium Climate Sensitivity in CESM2, Journal of Advances in Modeling Earth Systems, 14, 2022.

---

## Editor Decision (ED1)

**Minor comments on tc-2022-52**

- In the captions of Figures A1 and A2, add "Cross-hatched areas represent regions where this trend is significant (p < 0.05)".

- Lines 210-211: Aren't the estimates from Kuipers Munneke et al. (2012) based on regional model simulations? If so, I would not call them "observations". Even melt products derived from microwave data should probably not be called "observations".

- I suggest renaming section 3.4.1 "Comparison of the mean SMB with other products", or something similar pointing to the mean.

- L. 231-238 and 296-304: First, the word "signal" in this context is not very clear to me (also in other sentences throughout the manuscript). Is "xx% of the total SMB signal" equivalent to xx% of the total SMB? ("signal" may be used for trends or variances). Second, it is not clear to me how to calculate these percentages given that the SMB is the sum of positive (precip) and negative (sublimation, runoff) terms, so that I would expect e.g. +110% for precip and -10% for runoff for a total of 100%. This needs to be clarified in the revised manuscript.

- L. 274-276: "According to CESM2, increasing atmospheric temperatures throughout the 21st century are expected to increase precipitation across the AIS, and thus corresponding with future increases in AIS SMB". The end of this sentence does not sound good to me (note that I am not a native speaker).

- L. 282-287: $\frac{d\,SMB}{dT}$ should be $\frac{d\,\dot{SMB}}{d\,T}$ or $\frac{d}{dT}\frac{d\,SMB}{dt}$.

- Caption of Fig. 8: to make things clearer, you could specify "(left axis, solid)", "(right axis, dashed)".

- L. 399: "in the latest iteration of estimating future AIS contribution" was kind of correct in the previous version as it was implicitly pointing to ISMIP6 (Seroussi et al., 2020), but this is no longer meaningful for Siahaan et al. (2022) which is a single study.

---

## Author Response (AR2)

Minor comments on tc-2022-52

- In the captions of Figures A1 and A2, add "Cross-hatched areas represent regions where this trend is significant (p < 0.05)".

> We have added the suggested text to the captions of Figures A1 and A2.

- Lines 210-211: Aren't the estimates from Kuipers Munneke et al. (2012) based on regional model simulations? If so, I would not call them "observations". Even melt products derived from microwave data should probably not be called "observations".

> We have changed the sentence in line 211 to read:
>
> "Historical (1979-2015) surface melt in CESM2 has increased across much of the AIS (Fig. 5e), a trend that is absent from both regional climate model estimates of melt volume and microwave satellite observations of melt duration and area (Kuipers Munneke et al. 2012)."
>
> Kuipers Munneke et al. (2012) derive melt volumes from RACMO2 but also compare a RACMO-derived "cumulative melt surface" (the product of melt duration and melt area, CMS) with microwave satellite observations. In section 4 they describe a negative trend in both RACMO and observed CMS. Thus, we feel it is appropriate to mention that the positive melt trend in CESM2 is unmatched by regional climate model estimates *and* observations.

- I suggest renaming section 3.4.1 "Comparison of the mean SMB with other products", or something similar pointing to the mean.

> We have changed the name of section 3.4.1 to: "Comparison of the mean surface mass balance with other products".

- L. 231-238 and 296-304: First, the word "signal" in this context is not very clear to me (also in other sentences throughout the manuscript). Is "xx% of the total SMB signal" equivalent to xx% of the total SMB? ("signal" may be used for trends or variances). Second, it is not clear to me how to calculate these percentages given that the SMB is the sum of positive (precip) and negative (sublimation, runoff) terms, so that I would expect e.g. +110% for precip and -10% for runoff for a total of 100%. This needs to be clarified in the revised manuscript.

> Thanks for pointing out this confusion. To clarify, we have added a new section to the Methods:
>
> > **SMB component comparison**
> > To compare the relative importance of each SMB component during different time periods and from different model output we divided each component by the sum of the magnitude of all components, which we call the "SMB signal" throughout Section 3. For example, the contribution of runoff to the SMB signal was determined by:

$$runoff_{contribtion} = \frac{|runoff|}{|precipitation|+|evaporation/sublimation|+|runoff|},$$

where *precipitation* is the sum of both solid (snowfall) and liquid (rainfall) precipitation. This creates a standardized method to compare the relative importance of each SMB component among different models and scenarios.

We believe that this is the best way to compare the relative importance of each SMB component. If we maintain the positive contribution of precipitation and negative contribution of evaporation/sublimation and runoff, at the end of the SSP5-8.5 scenario we find that snowfall is +321% of the mean SMB and runoff is -304% of the mean SMB. From this it is not clear that the relative importance of precipitation is decreasing.

- L. 274-276: "According to CESM2, increasing atmospheric temperatures throughout the 21st century are expected to increase precipitation across the AIS, and thus corresponding with future increases in AIS SMB". The end of this sentence does not sound good to me (note that I am not a native speaker).

We have changed this sentence to: "According to CESM2, increasing atmospheric temperatures throughout the 21$^{st}$ century are expected to increase precipitation across the AIS, which will correspond with future increases in AIS SMB".

- L. 282-287: $\frac{d\,SMB}{dT}$ should be $\frac{d\,SMB}{d\,T}$ or $\frac{d}{dT}\frac{d\,SMB}{dt}$.

Because we are looking at the total change in SMB with respect to the total change in near surface air temperature over the 21$^{st}$ century (not taking a time derivative) we believe that the correct way to write this is: $\frac{\Delta SMB}{\Delta T}$. We have changed all instanced of $\frac{d\,SMB}{dT}$ to $\frac{\Delta SMB}{\Delta T}$ in lines 282-287.

We have also changed the phrase: "The rate of change in SMB with temperature ($\frac{d\,SMB}{dT}$)" to: "The 21$^{st}$ century change in SMB with respect to change in temperature ($\frac{\Delta SMB}{\Delta T}$)" to clarify this calculation.

- Caption of Fig. 8: to make things clearer, you could specify "(left axis, solid)", "(right axis, dashed)".

We have made this change in the caption of Fig. 8.

- L. 399: "in the latest iteration of estimating future AIS contribution" was kind of correct in the previous version as it was implicitly pointing to ISMIP6 (Seroussi et al., 2020), but this is no longer meaningful for Siahaan et al. (2022) which is a single study.

We have updated the first two sentences of the final paragraph (lines 367-369) to read:

"Recently, there has been some work done to couple ice sheet models and ESMs (Siaahan et al. 2021). However, even in the latest iteration of estimating future AIS contribution to sea level rise, Antarctic ice sheet models are largely simulated as a stand-alone, meaning they require climate forcing (Seroussi et al. 2020)."

---

## Editor Decision (ED2)

**Corrections before publication ( tc-2022-52 )**

Thank you for having addressed my minor comments. Please make these two last corrections:

- I am glad that you have defined what the "SMB signal" is, i.e. the sum of the absolute values (magnitude) of precipitation, sublimation and runoff. To be consistent with this definition, please use a different word than "signal" in line 24 (Introduction).

- I still have a concern with the Δ terms in section 3.5: SMB is in Gt yr$^{-1}$ and T is in °C, so ΔSMB/ΔT should be in Gt yr$^{-1}$ °C$^{-1}$, not in Gt yr$^{-2}$ °C$^{-1}$ as written in the text. If you keep these units, then it should be written ΔSMB/(Δt ΔT) where Δt is 85 years (from 2005-2015 to 2090-2100).

---

## Author Response (AR3)

**Corrections before publication ( tc-2022-52 )**

Thank you for having addressed my minor comments. Please make these two last corrections:

- I am glad that you have defined what the "SMB signal" is, i.e. the sum of the absolute values (magnitude) of precipitation, sublimation and runoff. To be consistent with this definition, please use a different word than "signal" in line 24 (Introduction).

> In Line 24 we have changed the following text: "Precipitation dominates the AIS SMB signal" to "Precipitation is the dominant SMB component"

- I still have a concern with the $\Delta$ terms in section 3.5: SMB is in Gt yr$^{-1}$ and T is in °C, so $\Delta$SMB/ $\Delta$T should be in Gt yr$^{-1}$°C$^{-1}$, not in Gt yr$^{-2}$ °C$^{-1}$ as written in the text. If you keep these units, then it should be written $\Delta$SMB/($\Delta$t $\Delta$T) where $\Delta$t is 85 years (from 2005-2015 to 2090-2100).

> We believe that the $\Delta$SMB/ $\Delta$T is the clearest way to define this term and so we have changed the units from Gt yr$^{-2}$ °C$^{-1}$ to Gt yr$^{-1}$°C$^{-1}$ in section 3.5.